# Uncertainty Analysis of Fluorescence-Based Oil-In-Water Monitors for Oil and Gas Produced Water

**DOI:** 10.3390/s20164435

**Published:** 2020-08-08

**Authors:** Dennis Severin Hansen, Stefan Jespersen, Mads Valentin Bram, Zhenyu Yang

**Affiliations:** Department of Energy Technology, Aalborg University, Niels Bohrs Vej 8, 6700 Esbjerg, Denmark; dsh@et.aau.dk (D.S.H.); sje@et.aau.dk (S.J.); mvb@et.aau.dk (M.V.B.)

**Keywords:** oil and gas industry, produced water, fluorescence-based monitor, oil-in-water concentration, real-time measurements, interferences, OLS, WLS, reproducibility, uncertainties

## Abstract

Offshore oil and gas facilities are currently measuring the oil-in-water (OiW) concentration in the produced water manually before discharging it into the ocean, which in most cases fulfills the government regulations. However, as stricter regulations and environmental concerns are increasing over time, the importance of measuring OiW in real-time intensifies. The significant amount of uncertainties associated with manual samplings, that is currently not taken into consideration, could potentially affect the acceptance of OiW monitors and lower the reputation of all online OiW measurement techniques. This work presents the performance of four fluorescence-based monitors on an in-house testing facility. Previous studies of a fluorescence-based monitor have raised concerns about the measurement of OiW concentration being flow-dependent. The proposed results show that the measurements from the fluorescence-based monitors are not or insignificantly flow-dependent. However, other parameters, such as gas bubbles and droplet sizes, do affect the measurement. Testing the monitors’ calibration method revealed that the weighted least square is preferred to achieve high reproducibility. Due to the high sensitivity to different compositions of atomic structures, other than aromatic hydrocarbons, the fluorescence-based monitor might not be feasible for measuring OiW concentrations in dynamic separation facilities with consistent changes. Nevertheless, they are still of interest for measuring the separation efficiency of a deoiling hydrocyclone to enhance its deoiling performance, as the separation efficiency is not dependent on OiW trueness but rather the OiW concentration before and after the hydrocyclone.

## 1. Introduction

The importance and awareness of accurate real-time measurements of oil-in-water (OiW) are increasing every year, due to the by-product of water production increases as a consequence of maturing offshore fields. According to literature, 70% of the world’s oil production is derived from mature fields [1,2]. As the general trend towards more sustainable production governed by discharge legislation imposing stricter policies and striving to achieve zero pollutant discharge, enhanced recovery techniques such as produced water re-injection (PWRI) have gained increasing attention to increase yield and reduce ocean discharge of oil [2,3,4]. Seen from an economic perspective, accurate information of oil and particle concentration, and the size distribution in the produced water (PW) for either re-injection or discharge, can be used for reporting, decision support, or even advanced control with the aim to achieve better operation in both separation and treatment [2].

PWRI is mainly deployed for reservoir pressure maintenance, which drives the reservoir production and increasing oil production by sweeping the reservoir [5,6]. In the Danish sector of the North Sea, ${}^{1}\;{/}_{3}$ of the PW is re-injected, which is around the average amount of PWRI for offshore processes in Europe (30%) [7,8,9]. Due to the increased environmental awareness and maturing fields, PWRI is required more often to reduce the emission of crude oil further, thereby oil will be present in the injection water (IW) [7]. This does not simply signify that the quality of the re-injected PW can be lowered. The presence of crude oil and other organic nutrients in PW can result in ideal conditions for bacterial growth, which may further lead to an increase of microbiologically influenced corrosion (MIC) [3]. Other common negative consequences, associated with crude oil in IW, occur due to the agglomeration-effect of oil on solid particles and the formation of asphaltene deposits. The injection of asphaltenes has shown to plug the porous media in the reservoir to such a degree that some fields in Saudi Arabia have prematurely stopped producing [10]. The flocculation of asphaltenes and the agglomeration effect of oil on solid particles have been recorded to increase the size of particles hundredfold, compared to what was expected downstream in the treatment process [11]. Therefore, whether PW is re-injected or discharged, it is essential to maintain a high PW quality for either optimizing the process or controlling the discharge concentration of oil.

Current offshore platforms rely on offline measurements of OiW concentration, but the reference methods vary due to different regulations worldwide, which renders comparison meaningless as measured OiW concentration is highly method-dependent [12,13,14]. The North Sea legislation (OSPAR) demands at least two manual samples each day using the gas chromatography-flame ionization detector (GC-FID) reference method [15]. OSPAR demands that the dispersed OiW concentration must be less than 30 mg/L, and the Environmental Protection Agency in Denmark further demands that the total amount of discharged oil in the Danish sector of the North Sea must be less than 222 tonnes annually [7,12,14,16]. The environmental risks associated with PW discharges will increase with increasing water production, especially as maturing fields have reached a water cut of 80% in 2015 [17]. For some of the more mature fields, the water cut even exceeds 90% [18,19]. This resulted in the former for Maersk Oil, which in 2015 discharged 193 of the allowed 202 tonnes [16]. Even though the current reference method (OSPAR GC-FID) has its limitations, a reference method is required for meaningful comparisons. Analyzing using the reference method is time-consuming, and the use of manual samplings is not sufficient for increasing production. The discharged OiW concentration between the two manual samples could exceed the regulations [12,14]. Since 2017, the Danish Environmental Protection Agency applied the following in Denmark through Executive Order no. 394 §9 [16].
“Online OiW monitors must be in operation and used for process optimization on the treatment plants for PW at all discharge sites …”“There must be continuous logging of the data, and the data must be stored for at least 5 years …”“Data collected with the online OiW monitors must be made available to the Danish Environmental Protection Agency if this is desired…”

Based on the Executive Order no. 394 §9 in Denmark, the oil and gas industry is consistently asked to invest in analytical methods that measure the discharge concentration of oil in PW. This emphasizes the importance of investigating online OiW monitors that could be a good candidate for measuring the same OiW concentration as the OSPAR GC-FID reference method. Continuous OiW measurement can be separated into two primary and two secondary purposes.

Primary purposes:Continual compliance with discharge legislation.Process optimization.

Secondary purposes:Data logging of PW quality for optimizing the re-injection process.Data logging for continuous revising of environmental legislation [9].

For measuring OiW concentration, there exist many commercially available online monitors with different measurement techniques designed for the oil and gas industry. However, to the authors’ knowledge, none of the non-reference methods have been robust and reliable enough to be accepted as a reference method for measuring OiW concentration. The online OiW monitors often require proper calibration and a non-hazard environment to be reliable. However, at in-field installations, large variations of mixtures, pressures, chemicals, and temperatures render a harsh and non-ideal environment for the monitors. Therefore, many cases where online OiW monitors have been installed without proper calibration and maintenance yielded doubtful and misleading measurements [20,21,22]. Even though most interferences may be slow and some close to constant, variations such as flow rate, pressure, OiW concentration, droplets/particle sizes, and particle concentration may change frequently. The irregularity of these variations completely depends on the installation, and it is well known that operation can change from one operator shift to another, which will impact the production [22]. Therefore, an online OiW monitor has to be able to maintain performance even when surrounding parameters change from its baseline calibration.

Based on promising results from previous investigations of a fluorescence-based monitor (Turner TD4100-XDC), this paper aims to examine the calibration method, their robustness to different interferences, and their reproducibility between each other. However, the OiW concentrations will not be validated by GC-FID in this paper.

### Previous Work Using the OiW Monitor

The Separation research group at Esbjerg campus has previously used the fluorescence-based OiW monitors in different investigations. However, several experiments have raised concern with regards to the monitors’ performance when exposed to certain interferences. The OiW monitors have been used to measure the separation efficiency of hydrocyclones and membranes, and different control methods manipulating different parts of the separation process on a pilot-plant at Aalborg University at Esbjerg campus, cf. [12,20,23]. Investigations of the previous papers did not raise any suspicion about the OiW monitors’ performance under different flow regimes as the measurements showed the expected outcome. Thus, at no given time was the true accuracy of the OiW concentration expected in the three papers. What especially raised concern was the discussion based on observed flow-dependency in the results by Bram et al. [24].

During the experiments, a significant correlation between the OiW concentration and the flow rate was observed, with OiW concentration measured on the sidestream and flow rate measured on the mainstream of the process. The outcome of the experiments showed that OiW concentration was proportional to the mainstream flow rate, which theoretically should not be true, as the two measurements are independent. Bram et al. [24] discuss whether it is the type and location of sampling point or interference on fluorescence intensity, e.g., gas bubbles, caused by flow-induced turbulence inside the sidestream configuration generated by shear forces.

Whether it is the sidestream sampling point, the separation process, or the instrument itself that causes the difference in OiW measurement will all be evaluated in this paper. The experiments executed on the pilot-plant will be presented in chronological order, too, when the experiments were executed. Experiments on different standalone systems are made independently of the experiments executed on the pilot-plant. The behavior observed by Bram et al. [24] will be recreated and repeated several times to clarify if it is a recurrent phenomenon.

The rest of the paper will discuss the comparison issues between non-reference methods (e.g., Turner 4100XDC) and reference methods (e.g., OSPAR GF-FID) in Section 2, and how that can be a constraint to the slow acceptance of new non-reference methods. Section 3 describes the different method and materials used for executing the experiments. Section 4 describes the experiment designs; Section 5 presents the results of the experiments. Section 6 discusses the outcome of the results, and last a conclusion is drawn in Section 7.

## 2. Non-Reference Methods Comparison with Reference Methods

For a non-reference method to fully be accepted as an alternative measurement technique in agreement with the OSPAR GC-FID method, it must first have high precision. Besides, it must be calibrated in order to increase trueness to gain high accuracy. The precision of an instrument indicates the uncertainty in the measurement and is referred to as the degree of repeatability and reproducibility in the measurement system [25,26]. Repeatability is the precision under the same conditions of method and equipment, used by the same operator to make measurements on the exact same sample. Reproducibility is the precision determined under the same conditions, but different pieces of equipment are used by different operators to make measurements on the exact same sample [27].

The measurement of OiW concentration is highly method-dependent, e.g., the OSPAR GC-FID method is limited to measuring the total hydrocarbons between $C_{7}$ and $C_{40}$ without toluene, ethylbenzene, and xylene (TEX), a modification of the ISO 9377-2 that is limited to measure the total hydrocarbons between $C_{10}$ and $C_{40}$. While the gravimetric-based method and the USA EPA method measure anything that is extractable by a solvent but not removed during a solvent evaporation process [14]. In addition, the measured quantities can vary under different conditions and even between different laboratories. Therefore, measuring OiW concentration is not only method-dependent but also procedure-dependent. As a result, a very detailed procedure for taking a sample, transportation, storage, and measuring in a laboratory is well described by the International Organization for Standardization (ISO). These detailed standards are necessary for valid comparison and reporting, especially if used to measure PW discharge. However, the method- and procedure-dependency to determine the “true” OiW concentration also introduces a substantial amount of uncertainties.

A cause-and-effect diagram is a common way to present the different contributions to the total measurement uncertainty [28]. The cause-and-effect diagram in Figure 1 shows the uncertainties related to determining the hydrocarbon oil index by following the ISO 9377-2 and the sampling procedure for crude oil in liquid ISO 3171 [29,30]. Thus, not the modified version from OSPAR. The GC-FID method calculates hydrocarbon oil index based on ISO 9377-2 [30]:(1)$$OiW=CF\frac{Vwf}{m_{1}-m_{2}},$$
where
(2)$$CF=\frac{A_{m}-b}{a}.$$

OiW is the hydrocarbon oil index [mg/L]; CF is the linear calibration curve, where *a* is the slope of the calibration curve; *b* is the intercept on the y-axis; $A_{m}$ is the integrated peak area measured of the sample extract; *V* is the volume of the final extract [mL]; $m_{1}$ is the mass of filled sampling bottle [g]; $m_{2}$ is the mass of the empty sample bottle [g]; *w* is the density of the water sample [g/mL]; *f* is any dilution factor of the sample extracted if necessary.

The measurement uncertainty ($u_{meas}$) can be divided into sampling ($u_{samp}$) and analysis ($u_{ana}$) uncertainties, visualized in Figure 1 with gray and orange, respectively. The parameters in Equation (Equation 1) are the representation of the main branches of the uncertainties related to the $u_{ana}$:(3)$$u_{meas}^{2}=u_{samp}^{2}+u_{ana}^{2}.$$

For a given OiW concentration of the sample, it is expected that the same analytical uncertainty applies to every measurement result [31]. This is not the case with sample uncertainty, as a greater part is derived from the heterogeneity of the target. The target is defined as the volume of produced water, at a particular time and location, that the sample intends to represent. By following the sampling procedure from ISO 3171, the PW sample should ideally be taken from rising flow in a vertical pipe section, with an isokinetic center lined pitot, in a turbulent region (RE ≳ 10,000) to ensure a representative well-mixed sample of the main flow, and then stored and transported onshore by following the ISO 9377-2 and ISO 5667-3, respectively. The uncertainties related to sampling are often dominant and occasionally may exceed 90% of the total measurement uncertainty [32]. Another author states that the sampling uncertainty may influence 75% of the total variance of measurement uncertainty [33].

Calculating or estimating the individual uncertainty distribution for each parameter (known as Type B uncertainty), as in Figure 1, can be a complicated procedure as it is difficult to determine all possible uncertainties. Therefore, ISO 21748 suggests using the reproducibility standard deviation ($S_{R}$) between laboratories as a representation of the measurement uncertainty (known as Type A uncertainty): $S_{R}≃u_{meas}$, but it may be an overestimation depending on the quality of the laboratory [25,28]. Thus, a measurement uncertainty can only give a representative result as good as the sample that is provided from the sample target [14,31]. As inter-laboratory reproducibility values are measured on the same sample, it does not account for the uncertainty related to the target. Therefore, uncertainties associated with the sample target are as least as important to address as the uncertainties associated with the measurement. This sample should ideally have exactly the same OiW concentration as the target, but in reality, this is impossible for heterogeneous mixtures [31,32]. The combined or total uncertainty should then be described as:(4)$$u_{c}^{2}=u_{target}^{2}+u_{meas}^{2}.$$

The uncertainty’s significance, related to the sample target, when measuring OiW concentrations is highly dependent on the location of the sampling point, sampling extraction process, the mixture of the heterogeneous solution, timing of sampling, etc. A study of environmental systems even shows that uncertainties between-operator and between-protocol are often much smaller than those caused by heterogeneity [32].

To measure the uncertainty related to the target for OiW concentration in an offshore oil and gas process, it must follow the sampling criteria from ISO 3171 to minimize the uncertainty related to heterogeneity. Then, to ideally measure the target uncertainty at more than one sampling point, at the exact same cross-sectional area of the pipe, could be utilized. The uncertainty between the extracted samples in a dynamic process could then be examined. However, for simplicity, the target uncertainty is always neglected, as most processes only have one sampling point at the same cross-section area of the pipe. Thus, if an online OiW monitor should be fairly tested against the reference method, the exact same sample must be used for both validation methods, online and manually, by sharing the target.

If assuming that the combined uncertainty can be expressed by the reproducibility, and the sample is a perfect representation of the body ($u_{target}=0$), the uncertainty using the reference method can intuitively be calculated from the data given in ISO 9377-2, as seen in Table 1. The reproducibility variance is assumed to be heteroskedastic, i.e., as OiW concentration increases, the variance increases with the same percentage for the entire range of OiW concentration of interest.

The variables in Table 1 are as follows. *L* is the number of laboratories after exclusion of outliers; *n* is the number of results after exclusion of outliers; $\bar{\bar{x}}$ is the grand mean of all results free from outliers; $x_{soll}$ is the true value; $S_{R}$ is the reproducibility standard deviation; $CV_{R}$ is the reproducibility coefficient of the sum of repeatability within-laboratories and between-laboratory variance; $S_{r}$ is the repeatability standard deviation; $CV_{r}$ is the repeatability coefficient of variation within-laboratory consistency [30].

If considering that an extracted sample from an offshore oil-producing platform has a “true” OiW concentration of 20 mg/L under ideal homogenous conditions, then
(5)$$u_{target}=0,$$
where the only uncertainties related to the true value is the reproducibility as a representation of the measurement uncertainty,
(6)$$u_{meas}=CV_{R}\bar{\bar{x}}.$$

As $CV_{R}$ varies randomly for each $\bar{\bar{x}}$ concentration, $u_{meas}$ is calculated as examples for best and worst case of an extracted sample of x= 20 mg/L:(7)$$u_{meas_{best}}=CV_{R_{1}}x=1.92$$
and
(8)$$u_{meas_{worst}}=CV_{R_{4}}x=8.1.$$

A combined standard uncertainty ($u_{c}$) that accounts for contributions from all important uncertainty components, in these particular examples $u_{meas_{best}}$ or $u_{meas_{worst}}$ as $u_{target}=0$ is
(9)$$u_{c}=\sqrt{u_{meas}^{2}+u_{target}^{2}}.$$

As $u_{c}$’s probability is only 68%, it is necessary to express it as expanded uncertainty (*U*) with a coverage factor (*k*). In these examples, a normal distribution with 95% confidence interval is considered (k=1.96):(10)$$U_{best}=u_{c_{best}}k=3.76$$
and
(11)$$U_{worst}=u_{c_{worst}}k=15.88.$$
For these particular examples, the OiW concentration should be described as:(12)$$OiW_{best}=20\pm 3.76,k=1.96,norm$$
and
(13)$$OiW_{worst}=20\pm 15.88,k=1.96,norm.$$

By considering those examples, it is within 95% confidence that the OiW in the best case does not exceed the discharge limit of 30 mg/L. For $OiW_{worst}$, the probability of measuring above 30 mg/L with a true solution of 20 mg/L is ≈5.5%, without even including the uncertainties related to the sampling target.

Due to the significant amount of uncertainties associated with the GC-FID method, the “error” of the measured values from an online monitor could potentially be a result of the GC-FID method rather than the performance of the online monitor. In worst case, an acceptable online monitor could be rejected despite being truly accurate and may have a negative influence on the reputation of all online OiW measurements. However, reaching a good understanding of the uncertainties associated with the reference method could facilitate the industrial acceptance of online OiW monitors [9].

## 3. Materials and Methods

This section describes the investigated four OiW monitors in this paper and the experimental setups used for validating the performance of the OiW monitors under different operating conditions.

### 3.1. Setup and Calibration of the OiW Monitors

The Turner TD-4100XDC online monitor detects the content of aromatic hydrocarbons in PW using fluorometry. The relative fluorescence units (RFU) are then converted to parts per million (ppm) through a calibration curve. As the OiW monitors are very sensitive to different compositions of atomic structures in its spectral excitation region, it must be calibrated to the mixture of the oil composition. Based on previous results using ARDECA SAE 30 motor oil, a non-detergent SAE 30 motor oil from Midland has been selected instead, as it is free from any surfactants dispersing and dissolving the OiW [34,35]. Even though the OiW monitors are able to detect dissolved oil, the separation process is unable to separate dissolved oil, and the separation efficiency will be inadequate. As the accuracy is nothing more than a comparison and adjustment of the relationship between the readings of the OiW monitors and the GC-FID method, the reproducibility between the four OiW monitors is investigated and not the OiW concentration according to the reference method. More detailed descriptions of the instrument and the calibration procedure are described in previous work [12,34,36]. The calibration of the OiW monitor follows the guideline (multipoint direct calibration) documented by the manufacturer, where at least two samples for a linear situation are recommended covering at least 50% of the desired monitoring range [37]. However, if the fluorescence of oil follows a nonlinear trend, multiple points are necessary. It should be evaluated whether a linear or nonlinear calibration equation produces the best calibration curve. The RFU sensitivity was adjusted by a physical sensitivity screw that sets the basic operating level. The aim of adjusting the operating sensitivity level was to secure the RFU response of all four OiW monitors are as similar as possible before calibration. 400 ppm was chosen as the highest value of interest, as the oil concentration is often less than 200 ppm, but noted to by another author to be below 400 ppm, before entering the deoiling hydrocylone as the second stage of the separation process [38,39].

The default identification from the manufacturer uses ordinary least square (OLS) regression to estimate the best-fit to a first- or second-order polynomial, and thereby implicitly assume homoskedastic random error (ϵ) by using unweighted regression calibrations. For OLS to be the best linear unbiased estimator (BLUE), five assumptions must be assumed according to the Gauss–Markov theorem: the true data trend is linear and that for all values of *x* the ϵ’s are independent, normally distributed with a mean of zero and have the same standard deviation throughout the region of interest (homoscedasticity) [40,41]. Although the first assumptions simplify the reality, proper calibration of the instrument might still achieve a close approximation to the true value. However, if heteroskedasticity is present, OLS loses its efficiency and is not BLUE anymore [41]. By looking at the box plot of the calibration data for all four OiW monitors in Figure 2, the combined 280 samples, 70 samples for each OiW monitor, clearly indicates that ϵ exhibit heteroskedasticity. The variance increases when the signal intensity increases, and that the RFU and ppm relation follows a linear trend.

Ignoring the fact that nonconstant variance might result in suboptimal calibration and invalid uncertainty estimates [41,42], instead, weighted least square (WLS) could be used as it may not be reasonable to assume that every calibration concentration should be treated equally. One disadvantage of using WLS is the weighting is often unknown, but an appropriate weighting factor ($w_{i}$) for WLS may well be the reciprocal of the variance ($\sigma^{-2}$) for each calibration concentration. This weighting will strengthen the influence on the parameter estimates the lower the variance is. The calibration function of interest for both OLS and WLS follows the linear function
(14)y=ax+b,
where *x* is the dependent concentration of OiW [RFU], *y* is the independent measurement unit of each the OiW monitors [ppm], *a* is the slope (sensitivity), and *b* is the intercept on the y-axis (offset). The two parameters *a* and *b* can be calculated by the regression formulas:(15)$$b=\frac{SS_{xy}}{SS_{xx}}$$
and
(16)$$a=\bar{y}-b\bar{x},$$
where $\bar{x}$ and $\bar{y}$ are the arithmetic weighted means of $x_{i}$ and $y_{i}$, respectively, for every *i* observations from 1 to *n*. $SS_{xx}$ and $SS_{xy}$ are the weighted sum of squares around the mean and the weighted cross-product of sum of squares for $x_{i}$ and $y_{i}$, respectively. For determining $\bar{x}$, $\bar{y}$, $SS_{xx}$, $SS_{xy}$, SSE, and $\hat{s}$, they are measured as in Equations (Equation 17)–(Equation 22), where the only difference between OLS and WLS is $w_{i}$. For WLS, $w_{i}=s_{i}^{-2}$, where $s_{i}=\sqrt{\frac{\sum_{i=1}^{n}e_{i}^{2}}{n-2}}$ and $e_{i}=y_{i}-{\hat{y}}_{i}$. “^∧^” denotes that it is predicted based on observed data. By simply making the weighting factor $w_{i}=1$, the OLS is equivalently achieved:(17)$$\bar{x}=\frac{\sum_{i=1}^{n}w_{i}x_{i}}{\sum_{i=1}^{n}w_{i}},$$
(18)$$\bar{y}=\frac{\sum_{i=1}^{n}w_{i}y_{i}}{\sum_{i=1}^{n}w_{i}},$$
(19)$$SS_{xx}=\sum_{i=1}^{n}w_{i}{\left(x_{i}-\bar{x}\right)}^{2},$$
(20)$$SS_{xy}=\sum_{i=1}^{n}w_{i}\left(x_{i}-\bar{x}\right)\left(y_{i}-\bar{y}\right),$$
(21)$$SSE=\sum_{i=1}^{n}w_{i}{\left(y_{i}-\hat{y}\right)}^{2},$$
and
(22)$$\hat{s}=\sqrt{\frac{SSE}{DOF}}.$$

As the estimation of the parameters for a linear regression *a* and *b* uses two degrees of freedom; DOF=n−2. The confidence interval (CI) and prediction interval (PI) are calculated, respectively, where the subscript “0” indicates the set of values which falls within the CI and PI [40,41]:(23)$$CI=ax_{0}+b\pm t_{0.025}\hat{s}\sqrt{\frac{1}{n}+\frac{{\left(x_{0}-\bar{x}\right)}^{2}}{SS_{xx}}}$$
and
(24)$$PI=ax_{0}+b\pm t_{0.025}\hat{s}\sqrt{\frac{1}{w_{0}}+\frac{1}{n}+\frac{{\left(x_{0}-\bar{x}\right)}^{2}}{SS_{xx}}}.$$

The $t_{0.025}$ is the t-score with two-tailed 95% confidence of a t-distribution with DOF=68. The true linear regression line should, with 95% probability, lie within the CI calculated from the sample data, and the PI should be an estimation of which future observations will fall within 95% probability. The attentiveness of this subject is due to when heterogeneity of variance is present, a substantial impact on prediction and calibration may occur, which will affect the precision and accuracy of the OiW monitor. Figure 3 illustrates the linear regression for each of the four OiW monitors with a 95% CI bounded by the inner dashed lines, and a 95% PI bounded by the outer dashed lines for both OLS and WLS.

As the true variance throughout the calibration region of interest is unknown, $w_{0}$ is estimated based on the predicted variance function:(25)$$s\left(\hat{y}\right)=c_{1}\hat{y}+c_{0}$$
and
(26)$$w_{0}=s^{-2}\left(\hat{y}\right),$$
where $c_{0}$ and $c_{1}$ are obtained by OLS on known sample standard deviation ($s_{i}$) in $y_{i}$ = (0,10,20,50,100,150,300)ppm. Equation (Equation 25) may only be a suboptimal representation of the variance function for the OiW monitor. However, a large amount of calibration replicates would be needed to determine whether or not Equation (Equation 25) is the best variance function. Noblitt et al. [41] describe other candidates of variance functions, though they will not be part of this paper.

Whether OLS or WLS provides better predictions of the true OiW measurement will be evaluated in the experiment *“Performance evaluation of the four OiW monitors’ calibration procedure and the related uncertainties”.*

### 3.2. Experiment Setups

The setup, located at Aalborg University in Esbjerg, is mainly used to emulate the main parts of an offshore deoiling process to validate potential performance improvements of different applied control strategies. An overview of the main process parts is shown in Figure 4. The system has, in the past, been used for several investigations to test/develop new control algorithms for process systems, modeling, slugging, fault detection/diagnosis, instrumentation performance, and HMI design [13,43,44,45,46,47].

As the pilot-plant is a laboratory setup, it is equipped with an excess of sensors and actuators, even at locations that might be nonexisting at industrial plants, to gain knowledge of indeterminate flow states and conditions. In this investigation on the OiW monitors’ performance, only the support section and the hydrocyclone section are used, the rest are bypassed. A more detailed illustration of the components and instructions, used in this paper for executing the experiments on the pilot-plant, can be seen in Figure 5a and Table 2, respectively. Notice a T-junction was used for splitting the two-phase flow between sidestream and mainstream. The T-junction’s branch-arm has a ${}^{3}\;{/}_{8}$” diameter (1”=25.4
mm) leading the flow into the sidestream of the process, and the mainstream of the two-phase OiW flow continuing through the run-arm of the T-junction.

Three standalone setups have been constructed, as seen in Figure 5b–d, with the corresponding Table 3, to isolate different parameters that can affect OiW measurements, and to evaluate the performance of calibrating the OiW monitors based on OLS and WLS.

Addition to Figure 5a, the flow path with nominal diameter and length of pipes between instruments is going through $C_{1}$, $C_{2}$, and $Q_{s}$ on the sidestream, hydrocyclone, and end through $V_{u}$ before returning to the water/oil tank, are illustrated in Figure 6. This is particularly important to consider when analyzing the fluid mechanics of the pilot-plant and to evaluate if the OiW monitors are properly connected according to the specification from the manufacturer.

## 4. Experiment Design

This section will describe the design and objective of all executed experiments with the common aim to investigate various interferences that could affect the OiW monitor. Due to previous results, which showed how a varying $Q_{i}$ influences the OiW concentration measured by one OiW monitor on sidestream, see Bram et al. [24], it is of interest to investigate whether this phenomenon on the pilot-plant is a consistent trend, by replicating the experiment. To observe the performance of the OiW monitors and the pilot-plant design, the following experiments were designed and executed.
Experiment designs executed on the by-passed pilot-plant setup, see Figure 5a:
○$Q_{i}$’s influence on $C_{1}$ and $C_{2}$.○Direct flow through sidestream.○Constant $Q_{i}$ with varying $Q_{s}$.○Constant pump speed with varying $Q_{s}$.Experiment designs executed on different standalone systems, see Figure 5b–d:
○Gas bubbles’ influence on $C_{1}$.○Repeatability investigation of flow-dependency of $C_{1}$.○Performance evaluation of the four OiW monitors’ calibration procedure and the related uncertainties.

Following the manufacturer’s recommendations, the flow rate should be within 1–2 L/min. Therefore, flow rates of 1.1 L/min and 1.9 L/min were chosen as the minimum and maximum flow rate, respectively, throughout all experiments executed through the OiW monitors.

### 4.1. Experiment Designs Executed on the By-Passed Pilot-Plant Setup

Two OiW monitors in series located on the sidestream were used to measure the OiW concentration. For all five experiments on the pilot-plant, there was no air injected into the setup, and the two mixers rotating with a constant speed of ≈137rpm (maximum speed). The performance of the hydrocyclone was not to be evaluated in this paper. However, if the performance and robustness of the OiW monitors are high, it will be of interest to investigate the optimal process conditions for the separation process of the hydrocyclone to lower the amount of discharge or increase production due to lower OiW concentration.

**$Q_{i}$’s influence on $C_{1}$ and $C_{2}$**: Investigate the observation observed by Bram et al. [24], to validate the OiW monitors’ flow-dependency in the pilot-plant.
$Q_{i}$ was stepped two times between 0.1 L/s and 0.4 L/s, and last time between 0.1 L/s and 0.5 L/s. This was accomplished using $Q_{i}$ as feedback to a PI-controller for controlling the rotational speed of CP.$Q_{s}$ is kept constant at 1.1 L/min by using $Q_{s}$ as feedback to a PI-controller for controlling $V_{s}$ opening degree.$V_{u}$ and $V_{o}$ were fully open throughout the experiment.


**Constant $Q_{i}$ with varying $Q_{s}$**: Isolate the varying flow rate by only affecting the sidestream, like the experiment with direct flow through sidestream. This was executed by varying $Q_{s}$ while compensating for having a constant $Q_{i}$. Note that this is only valid if the heterogeneous content is adequately mixed, and the pilot-plant physical design does not have any effect on the OiW mixture.
$Q_{i}$ was kept constant at 0.4 L/s by controlling the rotational speed of CP.$Q_{s}$ was stepped from 1.1 L/min to 1.9 L/min and back again by manipulating the $V_{s}$ opening degree.$V_{u}$ and $V_{o}$ was fully open throughout the experiment.

**Constant pump speed with varying $Q_{s}$**: Executed to investigate the pilot-plant’s design effects on the measurement. This was done by keeping the pump speed constant, meanwhile keeping $Q_{i}$ constant. Ideally, all parameters that would affect the OiW monitors will be at a constant level, except the uncertainty related to the heterogeneous mixture. However, as uncertainty related to the heterogeneous mixture is assumed normally distributed, the mean of the OiW concentration should be constant, even when $Q_{s}$ is changing. If not, the OiW monitors’ flow-dependency must be questioned and taken into consideration.CP was kept at a constant 90% pump speed.$Q_{s}$ was stepped from 1.1 L/min to 1.9 L/min and back again by manipulating the opening degree of $V_{s}$.$Q_{i}$ is kept constant at 0.4 L/s by controlling $V_{u}$’s opening degree.$V_{o}$ was fully closed throughout the experiment.


### 4.2. Experiment Designs Executed on Different Standalone Systems

To isolate the OiW monitors from the pilot-plant, different standalone systems were constructed to investigate different parameters that were suspected to influence the measurement of the OiW monitors.

**Gas bubbles’ influence on $C_{1}$**: Like turbidity monitors, the OiW monitors were suspected difficulties with interference from gas bubbles. For turbidity monitors, air bubbles are known to cause a false high turbidity reading when measuring the amount of light scattered [48]. However, results from another report testing a fluorescence-based monitor indicate that gas bubbles significantly reduce the measurement, as the gas bubbles potentially reduce the strength of both excitation light and fluorescence [22]. The change in fluorescence intensity occurs as gas bubbles scatter the light that distorts the fluorescence spectrum [49]. The setup used for detecting gas bubbles’ influence on $C_{1}$ can be seen in Figure 5b. The conditions of the experiment were as follows.

Fixed GP speed to have a constant flow rate of ≈1.1 L/min.Constant stirring speed of the magnetic stirrer.Constant air flow rate introduced down into the buffer tank, creating different sizes of air bubbles, together with the mixing behavior from the magnetic stirrer.

**Repeatability investigation of flow-dependency on $C_{1}$**: As entrained-air was observed to have an influence on the OiW measurement in the previous experiment, the pump and mixer were suspected of introducing unwanted bubbles into the system. This could be due to the motion of the mixer, inlet suction of the pump, or leakage. Increasing the flow rate also increased the number of bubbles observed in the outlet, which intuitively could be the reason for an increasing OiW concentration and not the flow rate itself. To eliminate the influence of the pump, one OiW monitor was gravity-fed instead with only tap water or demineralized water. The gravity-fed setup is illustrated in Figure 5d.

Flow rate was constant ≈1.1 L/min and ≈1.7 L/min, respectively, by manipulating $V_{m}$’s opening degree.

**Performance evaluation of the four OiW monitors’ calibration procedure and the related uncertainties**: Even though different parameters were observed to influence the measurement of the OiW concentration, the OiW monitors could still be feasible to use in the pilot-plant, if the monitors are consistent in their measurements and between each other. Four OiW monitors are used in this experiment for comparison of the instrument, see Figure 5c.

Nine different concentrations are tested: 0, 5, 10, 20, 40, 80, 160, 320, and 400 ppm. Demineralized water is used with a solution of oil and isopropanol to reduce the uncertainty in the heterogeneous mixture of OiW significantly. Demineralized water was chosen as the laboratory tap water was observed to fluoresce and change on a day-to-day basis.1.9 L/min was kept constant by controlling $V_{m}$’s opening degree.Fixed CP speed of 74%.

## 5. Results

This section provides the results of which flow regimes influence the four fluorescence-based monitors’ OiW measurements. The calibration presented in Figure 3 was not added to these experiments as only the trend of the OiW monitors’ measurements were of interest under different flow regimes. The investigations were divided into experiments executed on the pilot-plant and experiments executed on the standalone systems, as described in Section 4.

### 5.1. Experiment Results Executed on the By-Passed Pilot-Plant Setup

Before each experiment on the pilot-plant, the mixers had a run time of >20 min to ensure the OiW volume in the supply tank was well mixed. Noise was observed in all experiments on the pilot-plant originating from the mixers in the supply tank. The first part of the execution time for each experiment are used to ensure that the system had reached nominal operation. The initialization time depends on the experiment and mainly ensures that the system is flushed sufficiently as contents of oil, algae, and particles tend to settle or stick to the pipeline inner wall when the system is not in operation. Each experiment’s execution time was >100 min, to obtain steady-state measurements from the OiW monitors between each change of flow rate. The results of each experiment will be described and discussed in this section, and Section 6 will provide a holistic discussion of each topic. 

**$Q_{i}$’s influence on $C_{1}$ and $C_{2}$**: According to Figure 7, there is an indication that the OiW monitors are flow-dependent in the pilot-plant. $Q_{i}$ was stepped two times between 0.1 L/s and 0.4 L/s, and one last time between 0.1 L/s and 0.5 L/s. $Q_{s}$ was kept constant at 1.1 L/min. Figure 7 shows the same positive correlation as Bram et al. [24], when $Q_{i}$ increases the RFU increases. As $Q_{i}$ was controlled by the CP, the output signal of the CP has a similar shape as $Q_{i}$.

In this experiment the increment of the two OiW monitors was $C_{1}\approx 3000\mathrm{RFU}$ and $C_{2}\approx 2600\mathrm{RFU}$, when $Q_{i}$ was increased from 0.1L/s to 0.4L/s. The last step from 0.1L/s to 0.5L/s does not appear to increase the RFU measurement compared to $Q_{i}=0.4\;L/s$. This could indicate that the oil and water are separating differently in the process, depending on the flow rate. Looking at Figure 6, the transportation delay of the OiW mixture before reaching the sample point is relatively long, which will be a problem if the flow rate is not highly turbulent through the entire system. The worst condition for achieving sufficient turbulence is when the superficial velocity is lowest. For that particular experiment on the pilot-plant, this occurred when $Q_{i}$ is 0.1L/s and the pipeline diameter (*D*) is 2":(27)$$A=\frac{\pi }{4}D^{2},$$
(28)$$v=\frac{Q}{A},$$
and
(29)$$Re=\frac{\rho vD}{\mu }≃2545,$$
where $A=0.002\;m^{2}$, v=0.05m/s, $\rho =998.2\;\mathrm{kg}/m^{3}$, and $\mu =0.001\;\mathrm{Ns}/m^{2}\;@\;20{\;}^{\circ }C$. As Re<10,000 (even close to laminar Re<2300) in some situations of the experiment, it could indicate that the oil experience significant gravitational separation throughout the pipelines. Thus, making the measurements from the OiW monitors unjustifiably flow-dependent. It will also explain why there is little to no difference in the RFU measurement when $Q_{i}=0.4\;L/s$ or $Q_{i}=0.5\;L/s$ as Re≃10,182 and Re≃12,727, respectively. This also concludes that it is not possible to determine whether the OiW monitors are flow-dependent based on this experiment. The high spikes of RFU observed when $Q_{i}$ is increased, could be due to the presences of dead volume in the sidestream that is pushed through the OiW monitors after increasing the flow rate of $Q_{i}$, and a small systematic peak when $Q_{i}$ is decreased.

**Constant $Q_{i}$ with varying $Q_{s}$**: Figure 8 shows results of keeping $Q_{i}$ constant at 0.4 L/s, ensuring that the flow through the entire part of the pilot-plant is highly turbulent, and stepping $Q_{s}$ between 1.1 L/min and 1.9 L/min. Even though at high turbulence, where $Q_{i}$ is kept constant, RFU and $Q_{s}$ are still positively correlated. The increment of RFU after the increase of $Q_{s}$ is, nevertheless, only 200 RFU compared to the results in Figure 7, that has a tenfold increment. This narrows the issue down to being either due to the OiW monitors are, in fact, flow-dependent, even within its operation range recommended by the manufacturer, the T-junction sample point which is not a standardized way of sampling by ISO 3171, or it could be due to gas bubbles’ interference.

The result of CP increases slowly through the experiment as $V_{s}$ is closing gradually in order to force more flow through the OiW monitors. The back-pressure is increased due to the decreasing opening degree of $V_{s}$, which is the consequence of the increasing CP’s rotational speed. The steady increase of CP’s rotational speed does not appear to influence the RFU.

**Constant pump speed with varying $Q_{i}$**: Figure 9 is similar to Figure 8, but CP is kept constant at 90% rotational speed throughout the experiment, meanwhile controlling the varying $Q_{s}$ by manipulating the opening degree of $V_{s}$. However, the results in Figure 9 are distinguished from Figure 8 as no clear changes occurs when $Q_{s}$ is stepped from 1.1 L/min to 1.9 L/min and back again. This either indicates that increasing the rotational speed of CP decreases average droplet size by shear forces or increases entrained air and oil by having higher velocities of the carrier phase-flow near the water/oil tank’s outlet. To further investigate the OiW monitors, three different standalone systems were constructed to determine gas bubbles’ influence on the OiW monitors, the repeatability of each monitor under different flow conditions, and an extensive analysis of the calibration for the OiW monitors was carried out.

### 5.2. Experiment Results Executed on Standalone Systems

**Gas bubbles’ influence on $C_{1}$**: Figure 10 shows the results of introducing air bubbles into the standalone system Figure 5b. The solution used for this experiment only consists of tap water from the laboratory to isolate the heterogeneity of OiW, as it could be challenging to determine whether the difference in RFU was due to air introduction or due to a shift in the volume of oil droplets entering the OiW monitor. As presumed, a relatively high RFU was measured in the tap water by the OiW monitor, and the RFU value of the tap outlet was observed to change on a day-to-day basis. In this experiment, a mean of 1665 RFU was measured from 0 to 600 s when no air is added into the system. After 10 min, air was added into the system, as indicated in Figure 10 with a blue vertical line. Air pressure was applied until visible air bubbles were apparent in the buffer tank. The RFU increased and fluctuated during the injection of air, with a mean of 1783 RFU during 650–1200 s. The fluctuation is assumed as an outcome of the random sizes of air bubbles introduced by the air source passing the view cell. After ending the air injection, the RFU measurement slowly converged to its initial RFU value. However, the difference in mean before and after the addition of air is 14 RFU. That is likely due to the execution time of the experiment was not long enough for the added air to dissipate to the surrounding environment. It is noteworthy that when only air was present in the view cell, with some uncertainty to small micro/macro water droplets, the measurement was 80–90 RFU consistently. Nevertheless, it makes sense as the content of only air should ideally not fluoresce.

**Repeatability investigation of flow-dependency on $C_{1}$:** The results from gravity feeding one OiW monitor with both tap water and demineralized water with two different flow rates of 1.1 L/min and 1.7 L/min are seen in Figure 11. For experiments with tap water, the first three experiments with a flow rate of 1.7 L/min was rejected as the OiW concentration was consistently off compared to the other. The first three experiments inconsistencies were assumed happening due to start-up after a cleaning process. Demineralized water was selected as a reference, as ideally, it should not contain any matter capable of fluorescing. Tap water was known to fluoresce and was used for testing flow-dependency in another RFU range.

One-way analysis of variance (ANOVA) was carried out to test the null hypothesis ($H_{0}$) that differences in flow rate through the OiW monitor have equal mean RFU measurements. As for the F-statistic of ANOVA to be meaningful, it requires that the dependent variable is normally distributed in each group.

According to the one-way ANOVA test on the tap water with different flow rates, there is a 95.7% chance that the mean of RFU is equal at a flow rate of 1.7 L/min and 1.1 L/min. Thus, $H_{0}$ cannot be rejected. The boxplot in Figure 12 shows that the medians of the to flow rates are within the same range for tap water. Based on the results with different flow rates of tap water, if the $H_{0}$ is instead rejected, the flow-dependency is still insignificant compared to other variables’ influence on the measurements from the OiW monitors, as the difference in grand means and grand medians are 0.3 RFU and 4.0 RFU, respectively.

The one-way ANOVA test on the demineralized water with different flow rates shows that $H_{0}$ can be rejected and the boxplot shows similar result. Even though $H_{0}$ is rejected for demineralized water, the measurement grand mean and grand median is close to the result with tap water; 1.7 RFU and 2.1 RFU, respectively. The results from Figure 12 supports that the measurement from the OiW monitor on demineralized water is flow-dependent, though, it is insignificant. Interestingly, the experiments also show that the measurement of RFU is lower with a high flow rate of 1.7 L/min compared to a lower flow rate of 1.1 L/min. That is exactly the opposite as was observed in Figure 7 and Figure 8. Notice, measurement within 95% confidence interval are larger for tap water than demineralized water: (30)$$\bar{\bar{x}}\pm 11.64@\mathrm{tap}\mathrm{water},1.7\;L/\min,$$
(31)$$\bar{\bar{x}}\pm 8.24@\mathrm{tap}\mathrm{water},1.1\;L/\min,$$
(32)$$\bar{\bar{x}}\pm 0.51@\mathrm{demi}\mathrm{water},1.7\;L/\min,$$
and
(33)$$\bar{\bar{x}}\pm 0.38@\mathrm{demi}\mathrm{water},1.1\;L/\min,$$
where $\bar{\bar{x}}$ denotes the grand mean as reference. The big difference in 95% confidence interval between tap water and demineralized water is most likely due to fluorescence substances in the tap water compared to the demineralized water, that ideally should not contain any fluorescence substances.

**Performance evaluation of the four OiW monitors’ calibration procedure and the related uncertainties**: As mentioned in Section 1 and Section 2, having a good understanding of the uncertainties associated with the online monitors would help their industrial acceptance. However, for them to be accepted, proper calibration is vitally important to ensure trustworthy data. A 100 min experiment with eight different OiW concentrations were added throughout the experiment: 5, 10, 20, 40, 80, 160, 320, and 400 ppm starting at 0 ppm, as seen in Figure 13. All equipment was thoroughly cleaned before running the experiment to reduce the presence of contamination. Even though the accepted true value of the OiW concentration cannot be guaranteed without the use of the GC-FID reference method, the concentration uncertainties related to preparing the samples can be estimated based on Type B uncertainties, which will be presented later in this subsection. By analyzing the settling time for each injection of a new OiW concentration in Figure 13, it takes less than three minutes for the OiW concentration to settle in the system. The exact injection times are highlighted in Figure 13.

Based on the precision data presented in Table 1, the repeatability and reproducibility of the GC-FID method were calculated based on four different concentrations, in several different laboratories, with *n* number of samples for each OiW concentration. The repeatability and reproducibility followed the guidelines provided by ISO 21748 [25,50]. The same procedure was executed on the calibration data in Figure 3, to estimate the reproducibility as representation of the measurement uncertainty for both OLS and WLS, see Table 4 and Table 5, respectively. The Grubbs criterion was used for examining outliers, and the hypothesis of no outliers is rejected if
(34)$$G>\frac{n-1}{\sqrt{n}}\sqrt{\frac{t_{a/(2n),n-2}^{2}}{n-2+t_{a/(2n),n-2}^{2}}},$$
where *G* is the Grubbs criterion.

The $CV_{R}$’s can be added to the calibration measurements with an expanded uncertainty of coverage factor k=1.96 as in the previous Section 2. Based on the results in Table 4 and Table 5, a rough estimate of $CV_{R}$ = 10% was made for both OLS and WLS. Note that the reproducibility estimation is closely related to the repeatability for both OLS and WLS. This is highly due to not fulfilling all conditions for measuring the reproducibility stated in ISO 21748 [25]. The OiW measurement, on the same measurand, is done by different individuals but at the same time and in the same laboratory, which affects the results between-laboratory variance ($S_{L}^{2}$):
(35)$$S_{R}^{2}=S_{r}^{2}+S_{L}^{2}.$$

The prediction interval is another way to define the combined uncertainty of the OiW monitor. However, there is a chance that both PI and reproducibility, as a representation of combined uncertainty, are overestimated [28]. Even though overestimation often has its advantages for not exceeding the safety limit, it becomes difficult to predict the expected result and thereby misrepresents the ability to estimate an actual outcome, e.g., if the OiW monitors should be used as control feedback. Another way of predicting uncertainties is by calculating the type B uncertainties of the sampling and the type A uncertainty of the OiW monitors’ confidence interval. Like in Section 2, a cause-and-effect diagram is constructed to identify all relevant uncertainty sources. The parameters in the equation of measuring the OiW concentration are represented by the main branches in Figure 14. Although, it can be difficult to determine all related uncertainties. The cause-and-effect diagram in Figure 14 is divided into type B uncertainties (gray area), related to the manufactures of the different equipment stated uncertainties, and type A uncertainties (green area), related to the OiW monitors measured confidence interval from the calibration. The orange area in Figure 14 is related to sources of uncertainties due to the nature of the solution and human errors, which will not be included in the measurement of the combined uncertainty. This can be problematic as human errors always will be present, and the combined uncertainty might be underestimated. However, it will give a good indication of what the lowest amount of uncertainty would be based on the different equipment used for mixing the eight different solutions.

The different equipment used for preparing the different OiW concentrations can be seen in Table 6, and their additional systematic error and random error stated by the different manufacturers.

The standard uncertainty due to random error, u(V,rep), is often stated in a datasheet provided by the manufacturer of the pipette and implies the assumption of being normally distributed. For the different volumetric flasks, $VR_{1}$, $VR_{2}$, $VR_{3}$, and $VR_{4}$, the random error cannot, however, be determined by the manufacturer as it is not a mechanical error but rather based on the end-users ability to measure with the naked eye (human error). For simplicity, u(V,rep) of $VR_{1}$, $VR_{2}$, $VR_{3}$, and $VR_{4}$ are assumed zero. $VR_{5}$ and $VR_{6}$ follows a standard deviation of a normal distribution
(36)u(V,rep)=σ.

The standard uncertainty due to calibration or systematic error of a volume, u(V,cal), is commonly specified by the manufacturer on the equipment as $\pm u_{i}$. There is no information on the distribution or coverage factor of this uncertainty estimate, and the uncertainty is therefore conservatively assumed to be uniformly distributed. To convert the calibration uncertainty to standard uncertainty the calibration uncertainty is converted using
(37)$$u\left(V,cal\right)=\frac{u_{i}}{\sqrt{3}}.$$

Uncertainty, due to the temperature, affects the thermal expansion of both equipment and liquids. Although, the volumetric thermal expansion coefficient of Borosilicate glass is insignificant ($9.9\times 10^{-6}\;$°$C^{-1}$) and is therefore neglected. All the equipment is calibrated at 20 °C according to the manufacturers, and the laboratory’s temperature normally around 20±3°C, the uncertainty is conservatively assumed to be uniformly distributed as the deviation of temperature was not measured when the experiment was carried out:(38)$$u\left(V,temp\right)=\frac{V\Delta t\alpha_{V}}{\sqrt{3}},$$
where $\alpha_{V}$ is the volumetric thermal expansion coefficient of each different material (isopropanol, water, and oil), temperature error Δt=±3°C, and *V* is the specific volumetric measurement. For the design of the experiment 495mL isopropanol, 5mL oil, and 10 L of water was used. Uncertainties due to the temperature effects can then be calculated for water, isopropanol, oil, and as mixtures by using Equation (Equation 38).

To calculate the combined uncertainty for each concentration it is important to note that for every time the equipment is used, an amount of u(V,rep) and u(V,cal) is added to the total volume:(39)$$u\left(V\right)=\sqrt{u{\left(V,rep\right)}^{2}+u{\left(V,cal\right)}^{2}+u{\left(V,temp\right)}^{2}}$$

Table 7 shows the amount of equipment used for mixing the eight different concentrations from 5–400 ppm.

As a new concentration is added to the previous mixture of OiW, the total amount of mixture increases every time. This implies that the uncertainty from the previous mixture is added current one, see Table 8. The equations for estimating the predicted OiW concentration are as follows:(40)$$C_{stock}=\frac{V_{o_{tot}}}{V_{o_{tot}}+V_{iso_{tot}}}\times 10^{6}=\frac{V_{o_{tot}}}{V_{stock}}\times 10^{6}$$
and
(41)$$C_{i}\left(V_{i}\right)=\frac{V_{i}}{V_{w}+V_{i}}C_{stock}.$$

$C_{stock}$ is the 10,000 ppm stock solution of isopropanol and oil, $V_{o_{tot}}$ and $V_{iso_{tot}}$ are the total amount of oil and isopropanol added to the stock solution respectively. $C_{i}$ is the eight wanted OiW concentration that is aimed for, $V_{w}$ is the total amount of water used in the experiment, and $V_{i}$ is the different amounts of volumetric stock solution added to the experiment. Based on the standard uncertainties for each volumetric measurement of each equipment, the concentration uncertainty of the calibration can be determined by EURACHEM/CITAC [51]:(42)$$U_{cB}\left(C_{stock}\right)=C_{stock}\sqrt{{\left[\frac{u(V_{o_{tot}})}{V_{o_{tot}}}\right]}^{2}+{\left[\frac{u(V_{stock})}{V_{stock}}\right]}^{2}}$$
and
(43)$$U_{cB}\left(C_{i}\right)=C_{i}\sqrt{{\left[\frac{u(V_{w}+V_{i})}{V_{w}+V_{i}}\right]}^{2}+{\left[\frac{u(V_{i})}{V_{i}}\right]}^{2}+{\left[\frac{u(V_{o_{tot}})}{V_{o_{tot}}}\right]}^{2}+{\left[\frac{u(V_{stock})}{V_{stock}}\right]}^{2}},$$
where each OiW concentration’s uncertainties can be seen in Table 8. $u_{cB}$ is the type B uncertainties obtained from the uncertainty information given by the manufacturer of the used equipment, and $u_{cA}$ is the mean of all type A standard uncertainties obtained within one standard deviation (68% confidence interval) from the calibration of all four OiW monitors at the specific concentration, see Figure 3. Each OiW concentration’s uncertainties based on type A and type B can be seen in Table 8. Note that the previous $u_{cB}$, for each OiW concentration, is added to the continuous.

By JCGM [27] it is denoted that if the contributions to $u_{c}$ of the type A and type B standard uncertainties are denoted by $u_{cA}$ and $u_{cB}$, the combined uncertainty $u_{c}$ can be calculated as
(44)$$u_{c}=\sqrt{u_{cA}^{2}+u_{cB}^{2}},$$
and the expanded uncertainty, with 95% confidence interval (k=1.96) as
(45)$$U=u_{c}k$$

This is done for both OLS and WLS, as seen in Table 9.

Summarizing the three different methods of estimating the uncertainty related to preparing a sample and measuring the OiW concentration by the four OiW monitors:Using the prediction interval directly from the calibration of the OiW monitors.Estimating the reproducibility based on the calibration data.The estimated combined uncertainty based on type A and type B uncertainties.

Instead of looking at the RFU measurement of each OiW monitor in Figure 13, the OiW concentrations are measured by each OiW monitor for both OLS and WLS, which are shown in Figure 15 and Figure 16, respectively.

Only by looking at the two different ways of estimating the least square of the experiment in Figure 15 and Figure 16, there is a clear tendency that the deviation of WLS, in the lower region, is smaller than with OLS. At high concentrations, it is more difficult to determine which of the two ways deviates from the predicted concentration. This tendency also agrees with the theory that OLS tends to fit points that are at the upper calibration levels better than those points at the lower calibration levels. To evaluate the performance of OLS and WLS, an average for each OiW concentration is calculated based on the duration time between injection time minus the first three minutes listed in Table 10 and Table 11. As the accepted true OiW concentration cannot be given without the GC-FID reference method, the grand mean ($\bar{\bar{C}}$) of each OiW concentration based on all four monitors are calculated and their predicted OiW concentrations ($X_{pred}$) that were aimed for.

By analyzing the performance of OLS and WLS with the three different ways of determining uncertainties, a good indication of the OiW monitors’ reproducibility can be given. Figure 17 and Figure 18 show the results of the mean OiW concentrations from each OiW monitor for both OLS and WLS listed in Table 10 and Table 11, respectively. The $\bar{\bar{C}}$’s are used as a reference for each specific OiW concentration and both calibration methods, denoted as categories on the y-axes. The black squared marker in Figure 17 and Figure 18 represent the predicted OiW concentration for each target, and the numbered marker (①, ②, ③) represent the three different ways of determining the uncertainty.

By looking at the results for both OLS and WLS, in most cases, $X_{pred}$ lies within the measurements of the four OiW monitors. However, for the target at 40, 80, and 160 ppm solutions the OiW concentration of all four monitors were lower than the prediction aimed for with both calibration methods, which might be a result of insufficient amount of stock solution added to the setup.

Evaluating the performance of predicting the uncertainty related to the OiW monitors, each method’s performance is described for both calibration methods:


**OLS:**
①As the weighting factor within PI of OLS is equal to one, the result of using PI as uncertainty boundary are equal in all OiW concentration. Resulting in overestimation of uncertainty at a lower concentration, and might end in an underestimation at high OiW concentrations.②The 10% uncertainty estimation based on the reproducibility is applied, covering almost all OiW steady-state values the entire range except at 5 ppm. It is clearly the best way to represent the uncertainty related to OLS measurement compared to the other two methods.③The measurement of type B uncertainty was, as expected, difficult to include all uncertainties, resulted in an underestimation of the uncertainty above 40 ppm. The type A uncertainty from the CI of the OLS calibration is the main reason for the joined type A and type B uncertainty measurement fits within its boundary at the lower OiW concentration.



**WLS:**
①The weighting factor within PI of the WLS method is equal to the sample variance measured at each OiW concentration. The uncertainty estimation covers all OiW steady-state values in all ranges.②The same as for OLS, a 10% uncertainty estimation based on the reproducibility calculation is applied. The uncertainty range is lower than the PI but still cover all OiW steady-state values in the entire range.③As for OLS, the measurement of type B uncertainty for WLS was, as expected, also underestimated.


Even though the reproducibility may be a good representation of the uncertainty related to the OiW monitors for both calibration methods, it usually requires a substantial amount of inter-laboratory trials for calculating the reproducibility. Therefore, it may not be the best approach to achieve an uncertainty boundary related to the OiW concentration obtained from the OiW monitor. However, using the PI as uncertainty estimation is only valid for the WLS method due to the weighting factor.

## 6. Discussion

In general, RFU was used to represent the trend of OiW concentration, as the accepted true value of the fluid quantity was not able to be guaranteed on the pilot-plant. It will require a vast amount of cleaning to ensure that no previous dead volumes are present in the pilot-plant and all pipes are free of algae, bacteria, oil, grease, etc., all of which will influence the measurement. Even then, it can still not be guaranteed that contamination might inadvertently enter the system. Furthermore, samples of the OiW solution used in the pilot-plant should be analyzed by the GC-FID method to validate if the OiW monitors measure the accepted true value. As mentioned in Section 2 and Section 3.1, the most desirable outcome of the OiW monitors will be both high accuracy with high precision, but to achieve high accuracy with low uncertainty, it is necessary to have high precision. As directly investigating the OiW monitors’ precision can be challenging to accomplish with relatively high uncertainty in sampling, reproducibility was investigated to determine whether the OiW monitors can be used as a non-reference method. Figure 15 and Figure 16 show the result of calibrating the OiW monitors using OLS and WLS on the same data set. Based on the calibration data in Figure 3, there is a clear tendency of heteroskedasticity, where the variance increases when the signal intensity increases, and that the RFU and ppm relation follows a linear trend.

Looking at the results in Table 12 obtained from Figure 15 and Figure 16 of the two calibration methods, at no given time did the WLS have a larger deviation between the OiW monitors’ measurements than the OLS calibration method. Especially at the low OiW concentrations, the deviation of OLS was 2–3 times larger than with WLS. This indicates that WLS, as a calibration method, provides the higher reproducibility of the OiW monitors compared to OLS. Another analysis was to investigate if the uncertainty of the OiW monitors could be represented to an operator either for direct reporting of the OiW concentration with related uncertainty or as decision support. Including the uncertainty to the measurements will also fundamentally impact the control system design for expressing the discharge limit, if OiW monitors in the future should be used as control feedback for advanced control with the aim to achieve better operation in both the separation and treatment process. The results in Figure 18 show that PI and reproducibility are two good candidates for representing the uncertainty of OiW monitors’ measurements. However, reproducibility typically requires a substantial amount of inter-laboratory trials. Therefore, it is suggested that PI, which can mathematically be expressed based on calibration data, should represent the uncertainty of the OiW measurement, and reproducibility may be given as a reference to update the uncertainty parameter in the calibration curve. Note that the results from this experiment are taken under ideal conditions for the fluorescence-based monitors to investigate the calibration comparison, as isopropanol was added to the mixture for dissolving oil into the water phase.

During the experiments, variations of flow conditions caused the sidestream not to be a representation of the mainstream in the process. The results of Figure 7 raised the suspicion that the OiW monitors were flow-dependent, though further investigation confirmed that it was not the case. The results in Figure 11 and Figure 12, from the experiment on flow-dependency, showed an insignificant relationship between OiW concentration and flow rate based on the boxplot and the ANOVA tests, both with demineralized water and tap water. Demineralized water was selected as the basis of the reference as, ideally, it should not contain any fluorescence-sensitive matter. The tap water was known to fluoresce and was therefore used as a second RFU concentration for flow-dependency. Oil with emulsifiers could also have been used for this experiment to ensure that the results were not limited to water, but uncertainties related to homogeneity could increase due to oil, and isopropanol must be fully dissolved in the water phase. It was, therefore, debatable whether the extracted sidestream was representable due to operating conditions. By comparing Figure 7 and Figure 8, Reynolds number did have a huge impact on the measurement, and it is safe to conclude that the system cannot be operated below a mainstream of 0.4 L/s if turbulence is required throughout the system. It was furthermore questioned if the two-phases of the flow were uneven split in the T-junction due to hydrodynamics, and a phase maldistribution might have taken place between the run and the branch [52]. The T-junction sample point was also questioned to influence OiW concentration measurement, even when the flow is assumed highly turbulent; gravity, inertia, and pressure might still influence how the two-phase flow is divided between the run and branch of the T-junction [53]. Another drawback of using the T-junction as the sampling point is the propagation or amplification of other upstream processes effects that cause an uneven concentration split. An example of this could be the increasing shear forces of the CP that change the droplet size distribution, which should not ideally affect the OiW concentration. However, a shift in droplet sizes might affect the split between the run and the branch, which could potentially be the source of the observed increment in RFU as the flow rate is increased in Figure 8. This hypothesis is supported when comparing the results in Figure 8 and Figure 9, as the step input signal to $Q_{i}$ and $Q_{s}$ in both experiments are identical. By changing the controller and actuator in Figure 8 from controlling $Q_{i}$ by CP to controlling $Q_{i}$ by $V_{s}$, meanwhile keeping CP constant in Figure 9, the OiW measurement was kept constant. The dispersed liquid is likely more stable throughout the process when CP is kept constant, as increasing shear forces from the CP would probably decrease the average droplet size and thereby a better representation in the entire mainstream. Even though this is a positive effect for investigating the OiW monitors, it is undesirable as the average droplet size decrease, which heavily reduces the separation efficiency of the process [54]. The measurement of the OiW somewhat seems to be constant in Figure 9, but the system still has issues that can be addressed in the future to lower the uncertainty as much as possible related to sampling:Change the sample point from horizontal to be vertical.Use a sample probe for directing the rising flow through the sidestream.Use isokinetic sampling.Minimize the transport delay between the sample point and the OiW monitor as the manufacturer recommends a ${}^{1}\;{/}_{2}$" connection with a maximum flow rate of 2.0 L/min, which relates to transitional flow (Re=3358) and stratification can happen in the transport pipeline.

Air’s influence on the OiW monitors was also investigated on a standalone system. Figure 10 shows that the injection of air both introduces systematic and random error in the measurement of OiW. As the random error of air introduction only increases the uncertainty, seen as noise on the RFU signal, it might be acceptable if using a moving average of the OiW concentration, assuming zero-mean random noise, preferably including the measurements of the past minutes or more. Variations in the systematic error based on the volume and size distribution of gas bubbles will be a problem, as it will not be feasible accounting for air’s influence by simply recalibrating. However, seen from a normal oil separation process, systematic error due to gas bubbles might not be an issue as a three-phase separator will reduce the volume and size distribution of gas bubbles before the sample point. Further investigation should examine how the volume and size relationship of gas bubbles influences the OiW concentration.

Most of the discussion has focused on implementation and process criteria to reduce parameters that influence the OiW concentration, but it is not necessarily only a process design issue. The inner filter effect is a common internal issue of fluorescence-based monitors that describes the nonlinear relationship between the fluorescence intensity and the concentration of a fluorophore [55]. A decrease in fluorescence emission due to inner filter effect is caused by the absorption of the exciting light closer to the incident beam, which significantly diminishes the light intensity further away from it. Secondary inner filter effects are caused by reabsorption of fluorescence [55]. Inner filter effects could also have affected the results observed in Figure 8 and Figure 10. When the concentration is high, or the light absorption is high, the excitation beam is attenuated by the sample so that the surface perpendicular to the excitation beam fluoresces most powerful [56]. This effect might explain why the OiW monitors output a higher RFU even though the concentration is still the same, as smaller droplets that are generated by the high shear pump (CP) increases the oil droplets’ total surface area, which is at the current time under investigation.

Other effects that can influence the analysis are chemical quenchers, such as oxygen and chloride, where a quenching mechanism refers to decreases in the fluorescence intensity of a sample [57,58,59]. The most significant mechanism of oxygen quenchers is the collision of oxygen and phosphor molecules in the excited triplet state [56,57,60]. Even though chlorine has not been used in these experiments, it is necessary to address if the quenching mechanism happens when chlorine is present, as chlorine is often added to the process for killing microorganisms. In [58], the results showed that all polynuclear aromatic hydrocarbon (PAH) molecules decreased in fluorescence due to the presence of chlorine. Some of the PAH types did even show a 41% decrease in fluorescence due to chlorine [58]. Other more physical differences between the measurement of the OiW monitors could be due to differences in the photomultiplier or degradation of the fluorescent lamp over time. However, it should be possible to account for those by recalibration.

Based on the authors’ knowledge, one of the most significant issues with the use of fluorescence-based monitors is the big uncertainty related to what is fluorescing in the excitation region. Due to the high sensitivity to the presence of other atomic structures, in the same excitation region as aromatic hydrocarbons, the OiW monitors might not be feasible for measuring the exact OiW concentration in a highly dynamic separation facility with consistent changes of substances. As high reproducibility of the OiW monitors can be achieved with proper calibration, they can still be useful to determine the separation efficiency of separation equipment, such as deoiling hydrocylcones. As the separation efficiency is not dependent on OiW accuracy, but rather the ratio of OiW concentration across the hydrocyclone, the OiW monitors will still be feasible for enhancing the hydrocyclones deoiling performance. Another use of the OiW monitors could be in cooperation with other sensors, as a single monitor is unable to reduce uncertainty in its perception [61]. As uncertainty arises from various conditions for measuring the OiW concentration, it could be of interest to combine several sensors to gain more rich information. This can be done in several ways by using sensor fusion to increase the quality of data, increase reliability, or estimate unmeasured states. Redundancy of identical monitors would reduce the amount of uncertainty by averaging the value, seen in Figure 15 and Figure 16, and to compensate for sensor deprivation by fault-tolerant design [61]. However, it might not be the best solution as interference will affect all identical monitors, as discussed throughout the paper. Instead, a combination of different sensor types can be fused, i.e., Kalman filter, which provides a likelihood estimation of the measured OiW concentrations based on the different sensors [61]. As traditional Kalman filter requires the sensor measurement uncertainties, knowing the uncertainty of the OiW monitors will benefit the Kalman filter prediction [62]. This is usually executed by predicting the true value by producing estimates of the current state variables, along with their uncertainties [63]. The most weight is given to the value with the least uncertainty. As a result, the estimates produced by the Kalman filter tend to be closer to the true values than sensors used separately.

## 7. Conclusions

The paper presented an evaluation of four fluorescence-based monitors (Turner TD4100-XDC) that are sensitive to the content of aromatic oil in a mixture. The fluorescence-based monitor was thoroughly calibrated to a specific oil type used in all experiments. The “true” values of the fluorescence-based monitors, compared to the OSPAR GC-FID reference method, were at no given time measured, as the measurement of oil-in-water (OiW) is known to be highly methodology-dependent. Additionally, a relatively high amount of uncertainties are potentially associated with the reference method. From this assessment, the precision, repeatability, and reproducibility of the fluorescence-based monitors were investigated.

Testing the OiW monitors’ calibration method revealed that the weighted least square (WLS) is preferred to achieve higher reproducibility, compared to ordinary least square (OLS), due to the heteroskedastic behavior. Having a good understanding of the uncertainties associated with the OiW monitors will also help the industrial acceptance of including online monitors.

Previous studies of the fluorescence-based monitors raised concerns about the measurement of OiW concentration being flow-dependent. However, based on the results of the fluorescence-based monitors, they are not or at least insignificantly flow-dependent within its recommended flow rate range. The flow-dependency phenomenon of two fluorescence-based monitors, discussed by Bram et al. [24], was caused by insufficient representation of the process flow. To circumvent the transport stratification in the pipelines, the mainstream of the process must be operated at Re > 10,000. Furthermore, the horizontal used T-junction as a sampling point was questioned to split the two phases unevenly, and as a result, the testing facility is scheduled for replacement of the sampling point to further lower uncertainties.

The measurements of the OiW monitors were also observed and discussed to be affected by other interferences such as gas bubbles, droplet sizes, quenchers introduced by the solution’s composition, and presences of other atomic structures in the same excitation region. Due to the high sensitivity to different compositions of atomic structures other than aromatic hydrocarbons, the fluorescence-based monitors might not be feasible for measuring OiW concentrations in highly dynamic separation facilities with continuous changes of the fluid composition. Thus, different interference parameters influenced the measurements of the fluorescence-based monitors, they still have a high precision between each other, and it could still be of interest for measuring the separation efficiency of OiW separation processes downstream the gravity separator, such as deoiling hydrocyclones and membrane filtration systems, to enhance their deoiling performance. One advantage of measuring inlet and outlet OiW concentration of a separation process is that separation efficiency is a ratio of the two measurements, and therefore robust to interference that affects both of the monitors.

## Figures and Tables

**Figure 1 sensors-20-04435-f001:**
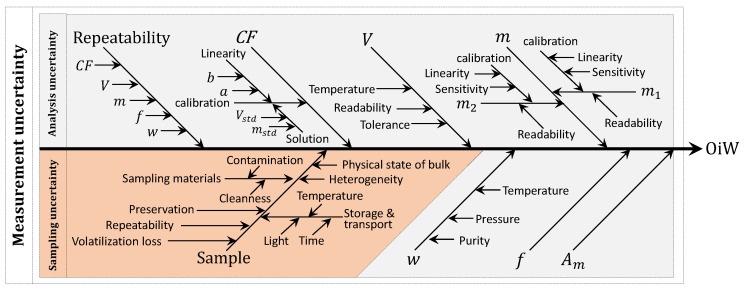
Cause-and-effect diagram of compounded uncertainties contribution to the measurement of OiW concentration by following the gas chromatography-flame ionization detector (GC-FID) reference method defined by OSPAR. The cause-and-effect diagram is divided into two sub-groups: Analysis uncertainty (gray) and sampling uncertainty (orange).

**Figure 2 sensors-20-04435-f002:**
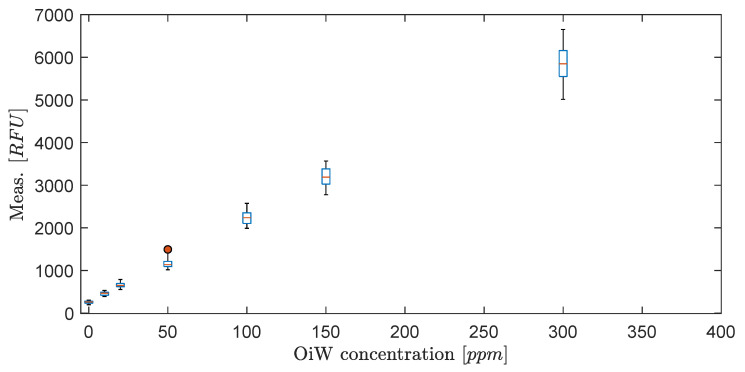
Combined box plot at each oil-in-water (OiW) concentration of all four OiW monitors’ calibration data. The calibration data consist of 280; 70 samples for each OiW monitor at following OiW concentrations: 0, 10, 20, 50 100, 150, 300 ppm.

**Figure 3 sensors-20-04435-f003:**
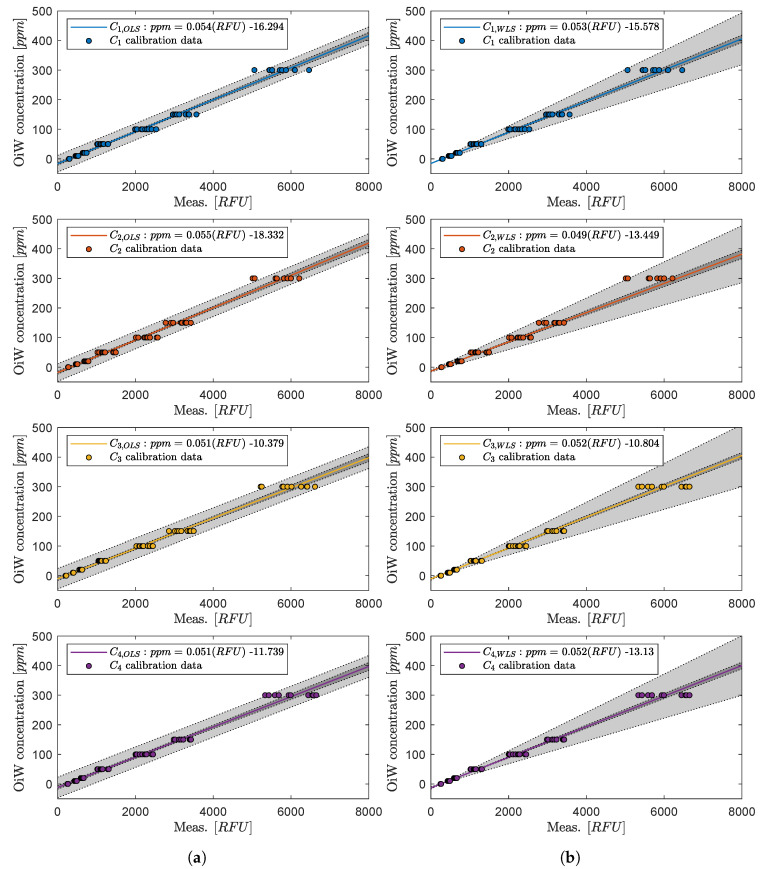
Parameter calibration for each of the four OiW monitors fitting to a linear regression: (**a**) illustrate the linear regression using OLS. (**b**) illustrate the linear regression using WLS.

**Figure 4 sensors-20-04435-f004:**
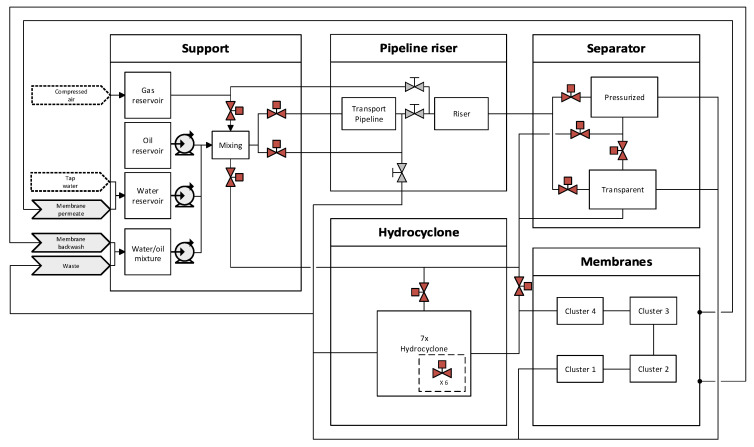
An overview of the bypass structure of the offshore pilot-plant at Aalborg University in Esbjerg.

**Figure 5 sensors-20-04435-f005:**
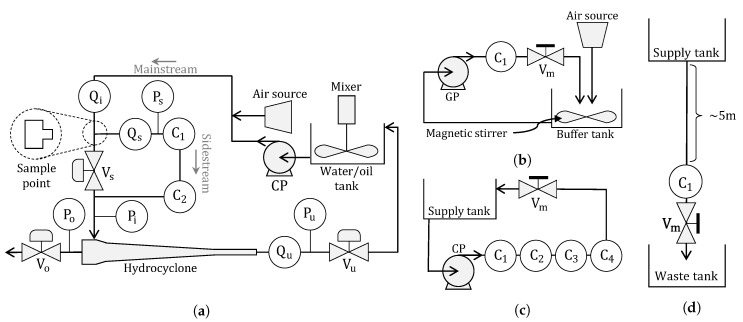
Piping and instrumentation diagram of the by-passed pilot-plant and three standalone setups: (**a**) illustrates the by-passed pilot-plant. (**b**) illustrates the setup for investigating gas bubbles’ influence on a OiW monitor. (**c**) illustrates the setup for comparison of the four OiW monitors’ calibration. (**d**) illustrates a gravitation feed setup to isolate internal influences.

**Figure 6 sensors-20-04435-f006:**
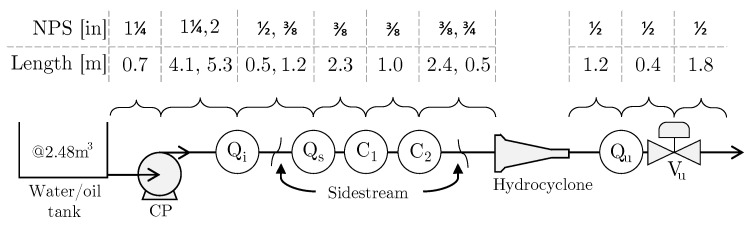
Illustration and table of the flow path with nominal diameter and length of pipes between instruments going through the sidestream.

**Figure 7 sensors-20-04435-f007:**
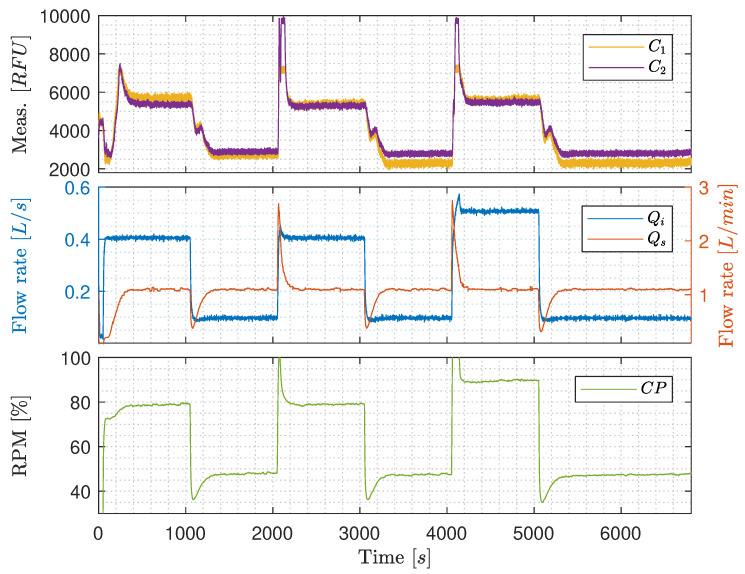
$Q_{i}$’s influence on $C_{1}$ and $C_{2}$.

**Figure 8 sensors-20-04435-f008:**
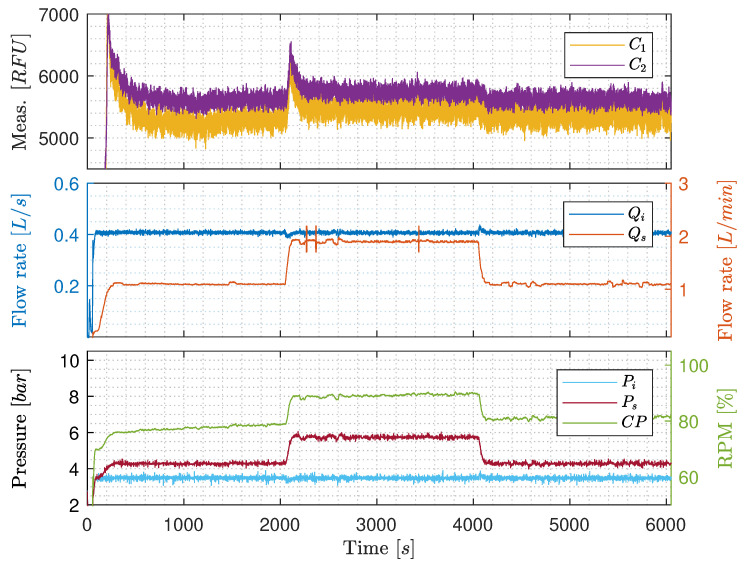
Constant $Q_{i}$ with varying $Q_{s}$.

**Figure 9 sensors-20-04435-f009:**
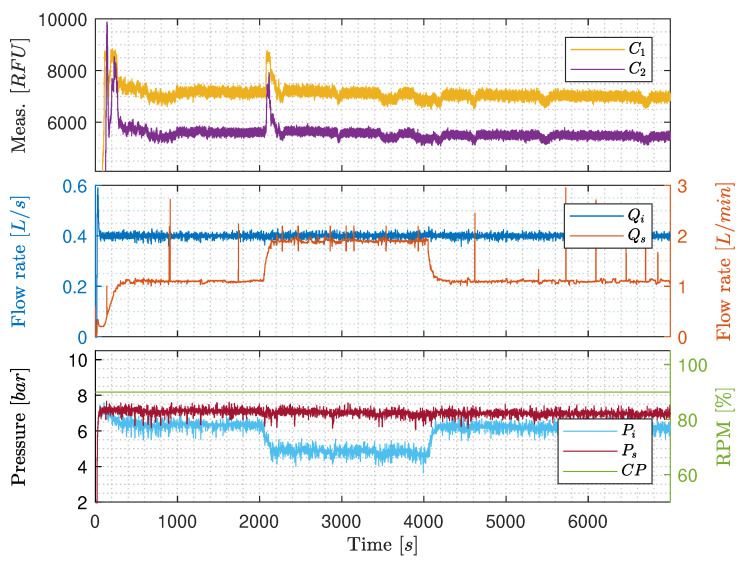
Constant pump speed with varying $Q_{s}$.

**Figure 10 sensors-20-04435-f010:**
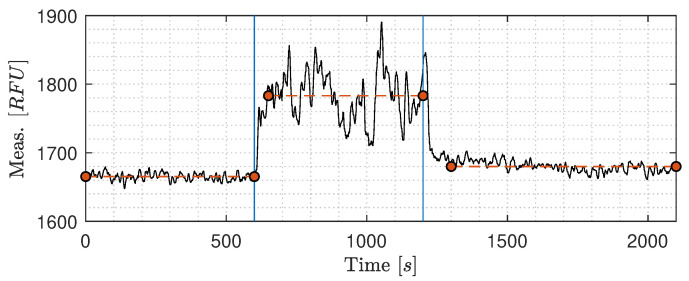
Gas bubbles’ influence on the OiW monitor.

**Figure 11 sensors-20-04435-f011:**
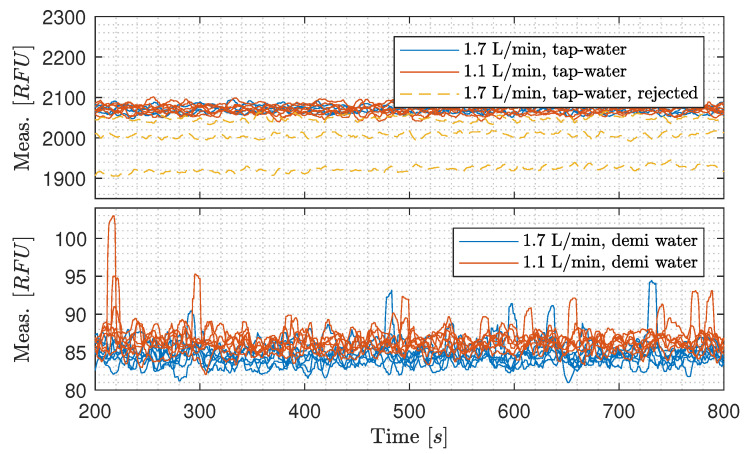
Repeatability investigation of flow-dependency on $C_{1}$ with demineralized water and tap water at two different flow rates.

**Figure 12 sensors-20-04435-f012:**
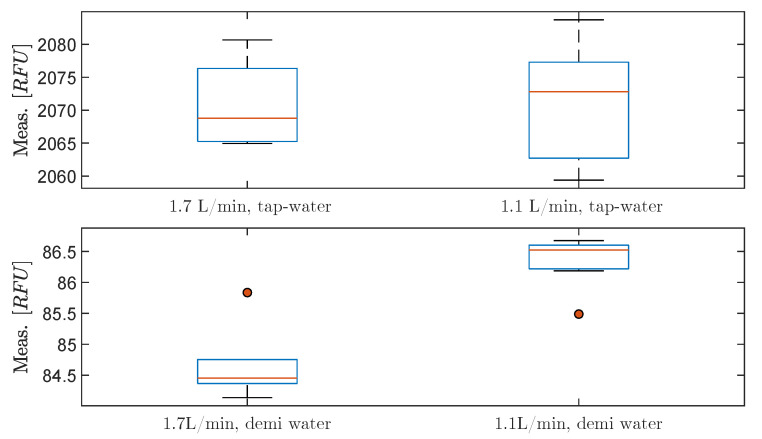
Boxplots for each type of water with different flow rates.

**Figure 13 sensors-20-04435-f013:**
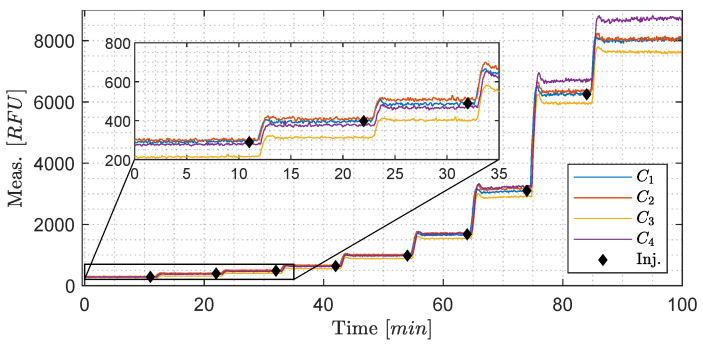
A 100 min experiment with eight different OiW concentrations were added throughout the experiment: 5, 10, 20, 40, 80, 160, 320, and 400 ppm starting at 0 ppm. The injection time is marked with black diamonds. The OiW measurements from the four OiW monitors are displayed in RFU, to evaluate the raw data of the OiW monitor before analyzing the calibration methods.

**Figure 14 sensors-20-04435-f014:**
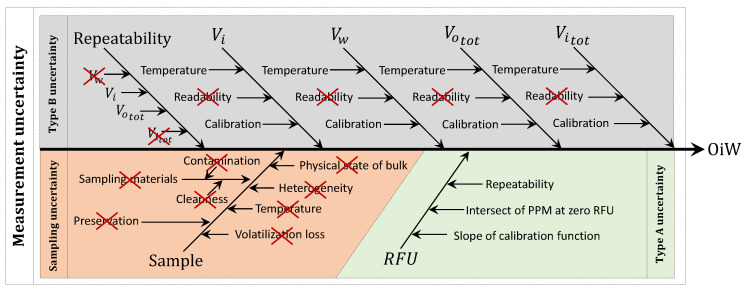
Cause-and-effect diagram of compounded uncertainties contribution to the measurement of OiW concentration by following the uncertainty related to each equipment used for mixing the eight different solutions. The cause-and-effect diagram is divided into three sub-groups: type B uncertainties (gray), type A uncertainties (green), and sampling uncertainties (orange).

**Figure 15 sensors-20-04435-f015:**
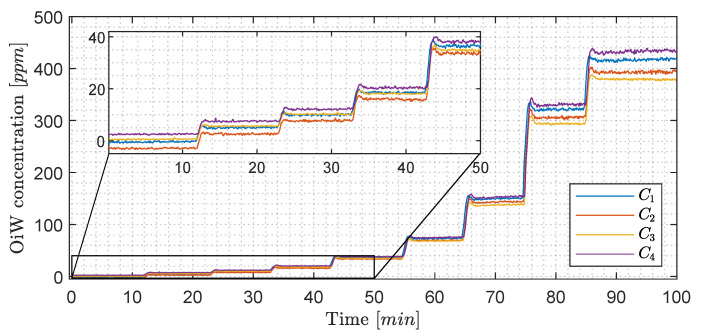
A 100 min experiment with eight different OiW concentrations as in Figure 13. The measurements from the four OiW monitors are displayed in ppm based on the OLS calibration method.

**Figure 16 sensors-20-04435-f016:**
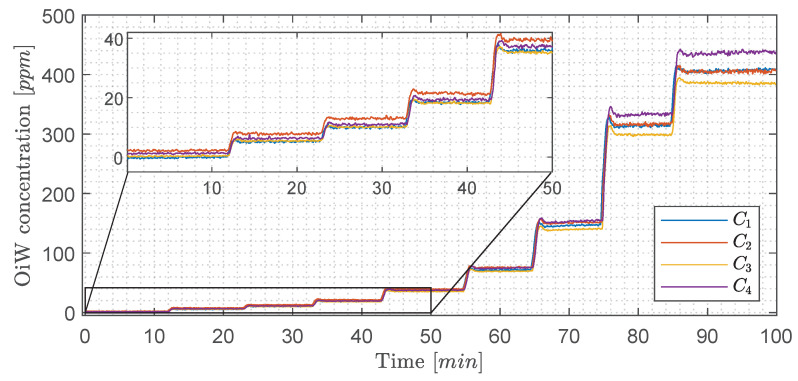
Similar as Figure 15; however, the measurements from the four OiW monitors are based on the WLS calibration method instead of OLS.

**Figure 17 sensors-20-04435-f017:**
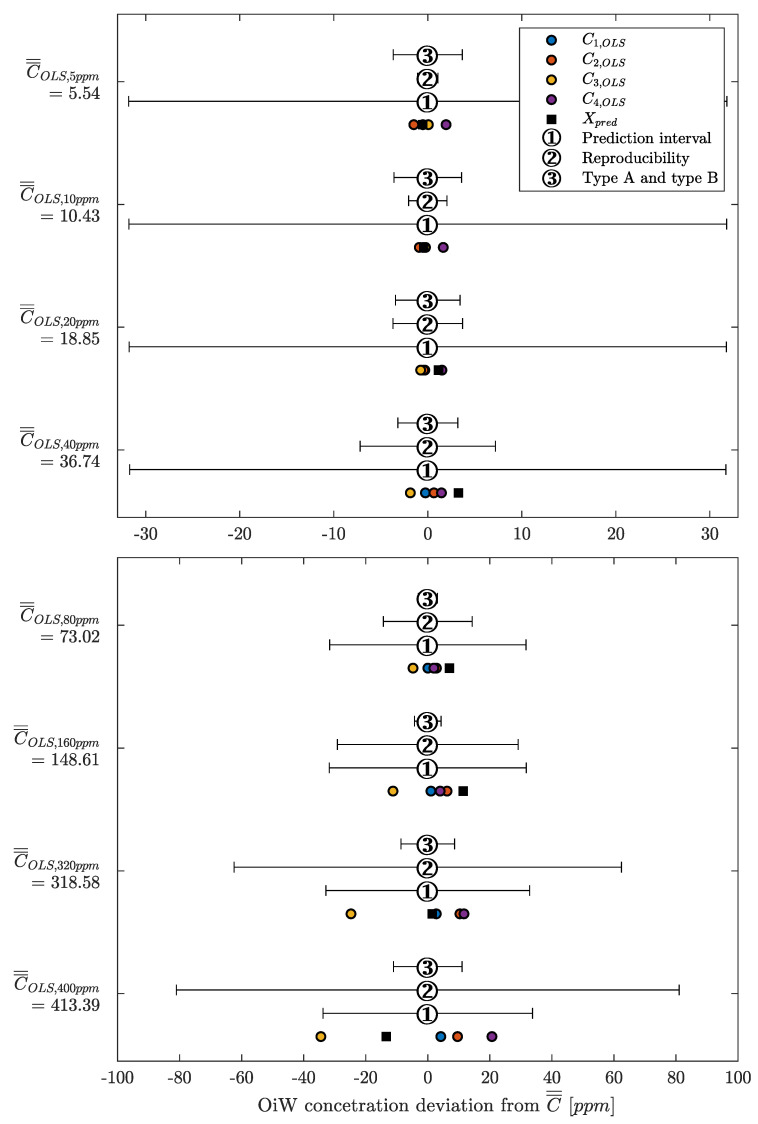
The analysis of three different methods for determining the best estimation of the OiW monitors’ uncertainty based on OLS in the entire range of interest. The grand mean for each concentration is used as a reference point and categorized on the y-axes. The plot is divided into two subplots for better visualization of the OiW concentration range of interest.

**Figure 18 sensors-20-04435-f018:**
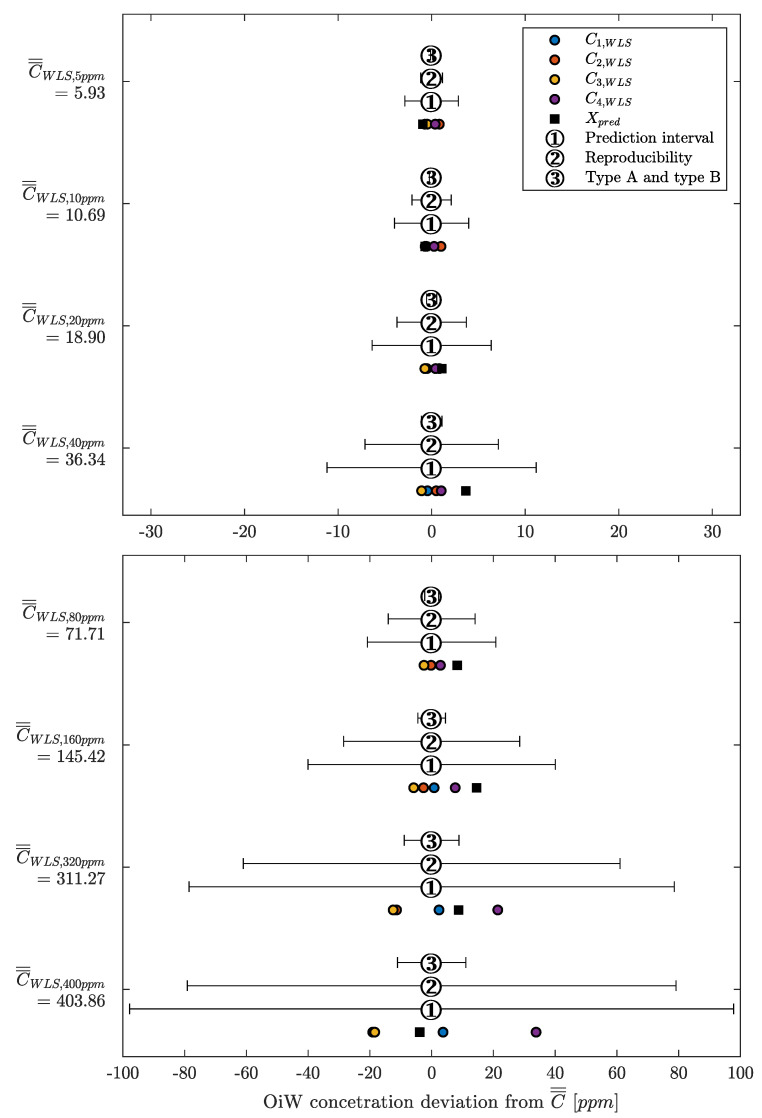
The analysis of three different methods for determining the best estimation of the OiW monitors’ uncertainty based on WLS in the entire range of interest. The grand mean for each concentration is used as a reference point and categorized on the y-axes. The plot is divided into two subplots for better scaling of the OiW concentration range of interest.

**Table 1 sensors-20-04435-t001:** Data of inter-laboratory trials published in ISO 9377-2 [30].

Sample No.	*L* [−]	*n* [−]	$\bar{\bar{x}}$ [mg/L]	$x_{\mathrm{soll}}$ [mg/L]	$S_{R}$ [mg/L]	${\mathrm{CV}}_{R}$ [%]	$S_{r}$ [mg/L]	${\mathrm{CV}}_{r}$ [%]
1	35	127	3.04	2.99	0.291	9.6	0.092	3.0
2	34	134	0.57	0.70	0.192	33.5	0.037	6.5
3	38	142	3.61	4.00	0.763	21.1	0.210	5.8
4	41	156	0.74	1.04	0.300	40.5	0.105	14.1

**Table 2 sensors-20-04435-t002:** Specifications of instruments used in the pilot-plant process. An illustration of the instruments’ position is seen in Figure 5a.

Component	Type	Description	Specifications
Water/oil tank	Custom made	Supply and waste tank for the oil and water mixture	2.48 $m^{3}$
Mixer	2 × Milton Roy Mixing VRP3051S90	Two mixers for mixing the immersible oil in water	$N_{max}$ = 137 rpm, 3 × 550 mm blades
CP	Grundfos CRNE5-9 A-P-G-V-HQQV	Centrifugal pump feeding the OiW separation system	6.9 $m^{3}/h$ at h = 68 m, $h_{max}$ = 93.3 m
Air source	-	Addition of air into the system if necessary	(1–7) barA
Hydrocyclone	Vortoil 35 mm liner	Single industrial cased hydrocyclone liner	-
$Q_{i}$ and $Q_{u}$	Bailey-Fischer-Porter 10DX4311C	Magnetic flowmeters measuring the inlet and underflow outlet of the hydrocyclone, respectively	0–1.64034 (0–1) L/s
$Q_{o}$	Emerson Micro Motion ELITE CMFS010M300N0ANACZZ	Coriolis flowmeter measuring the overflow outlet flowrate from the hydrocyclone	@0.002-97.0 kg/h
$P_{i}$, $P_{i}$, and $P_{o}$	Siemens Sitrans P200	Pressure transmitters measuring the pressure at their respective locations	(0–16) barA
$C_{1}$ and $C_{2}$	Turner-Design TD-4100XDC	Fluorescence-based OiW monitors measuring the OiW concentration before and after the hydrocyclone	(0–5000) ppm
$V_{s}$, $V_{o}$, $V_{u}$ and $V_{ub}$	Bürkert type 8802	Pneumatic continuous control valves controlling the flow in the system	Δt≲ 1 s, Hysteresis 1%, $P_{max}$ = 16 bar

**Table 3 sensors-20-04435-t003:** Specifications of instruments used for the standalone systems. An illustration of the instruments’ position is seen in Figure 6.

Component	Type	Description	Specifications
Buffer tank	VWR 213-1128	Supply and waste beaker with/without a magnetic stirrer for mixing the solution	1000mL
Supply tank	3H1/Y1.8/200	Supply plastic jerrycan gravity-feeding the OiW monitor	20 L
Waste tank	3H1/Y1.8/200	Plastic jerrycan for capturing the waste of the gravity-feeding system	20 L
GP	Greylor PQ-12	Gear pump supplied by a 0–24 V AC/DC power supply, feeding the standalone systems; Figure 5b	$Q_{max}=\;$0.132 $m^{3}/h$, $P_{max}\approx$2.4 bar
CP	Grundfos CRNE5-2 A-P-G-V-HQQV	Centrifugal pump feeding the standalone system; Figure 5c	6.9 $m^{3}/h$ at h = 12.9 m, $h_{max}=$ 20.6 m
Air source	-	Addition of air into the system	(1–7) barA
V	Swagelok	Ball valve manipulating the flow in the standalone systems	-
$C_{1}$, $C_{2}$, $C_{3}$, and $C_{4}$	Turner-Design TD-4100XDC	Fluorescence-based OiW monitor measuring the OiW concentration	(0–5000) ppm

**Table 4 sensors-20-04435-t004:** Data of inter-laboratory trials from the four OiW monitors based on ordinary least square (OLS) calibration.

Sample No.	*n* [−]	$\bar{\bar{x}}$ [ppm]	$x_{\mathrm{soll}}$ [ppm]	$S_{R}$ [ppm]	${\mathrm{CV}}_{R}$ [%]	$S_{r}$ [ppm]	${\mathrm{CV}}_{r}$ [%]
1	39	−0.54	0	2.11	−	0.25	−
2	39	10.13	10	1.68	16.58	0.94	9.27
3	37	20.58	20	1.42	6.93	1.26	6.16
4	38	47.27	50	5.99	12.78	5.92	12.60
5	40	103.76	100	8.97	8.64	8.89	8.57
6	38	154.53	150	9.80	6.34	9.48	6.13
7	40	294.26	300	24.50	8.33	23.39	7.95

**Table 5 sensors-20-04435-t005:** Data of inter-laboratory trials from the four OiW monitors based on weighted least square (WLS) calibration.

Sample No.	*n* [−]	$\bar{\bar{x}}$ [ppm]	$x_{\mathrm{soll}}$ [ppm]	$S_{R}$ [ppm]	${\mathrm{CV}}_{R}$ [%]	$S_{r}$ [ppm]	${\mathrm{CV}}_{r}$ [%]
1	39	0.04	0	0.26	−	0.24	−
2	39	10.42	10	0.96	9.23	0.91	8.77
3	37	20.52	20	1.91	9.31	1.19	5.80
4	38	46.20	50	5.71	12.37	5.54	11.98
5	40	101.73	100	8.66	8.51	8.62	8.47
6	38	151.57	150	10.28	6.78	9.32	6.15
7	40	287.92	300	26.42	9.18	23.01	7.99

**Table 6 sensors-20-04435-t006:** A list of the six different equipment used for preparing the samples of the experiment seen in Figure 13, and the corresponding systematic and random error.

Abbreviation	Volumetric Equipment	Volume	Systematic Error	Random Error
$VR_{1}$	Graduated cylinder, tall form, BLAUBRAND^®^, class A, 1000mL	1000mL	±5.00mL	−
$VR_{2}$	VWR^®^ Volumetric Flask, Class A, 500mL	500mL	±0.25mL	−
$VR_{3}$	Graduated cylinder, tall form, BLAUBRAND^®^, class A, 250mL	250mL	±1.00mL	−
$VR_{4}$	Graduated cylinder, tall form, BLAUBRAND^®^, class A, 100mL	100mL	±0.50mL	−
$VR_{5}$	Finnpipette^®^ F2: (0.5–5)mL	(0.5–5)mL	±40.0 µL	15.0 µL
$VR_{6}$	Gilson^™^ F148504: (10–100) µL	(10–100) µL	±1.5 µL	0.6 µL

**Table 7 sensors-20-04435-t007:** A list of used equipment used for mixing each concentration, $V_{i}-V_{i-1}$ represent the amount of stock solution that should be added to reach the target concentration.

$V_{w}$ [mL]	Used Equipment		
10,000	10 × $VR_{1}$		
**Wanted [ppm]**	**$V_{i}$ [mL]**	**$V_{i}-V_{i-1}$ [mL]**	**Used Equipment**
10,000 *	500	−	1 × $VR_{2}$, 1 × $VR_{5}$
0	0	0	−
5	5.00	5.00	1 × $VR_{5}$
10	10.01	5.01	1 × $VR_{5}$, 1 × $VR_{6}$
20	20.04	10.03	2 × $VR_{5}$, 1 × $VR_{6}$
40	40.16	20.12	4 × $VR_{5}$, 2 × $VR_{6}$
80	80.65	40.48	8 × $VR_{5}$, 5 × $VR_{6}$
160	162.60	81.96	1 × $VR_{4}$
320	330.58	167.96	1 × $VR_{3}$
400	416.67	86.09	1 × $VR_{4}$

* Stock solution of isopropanol and oil ($C_{stock}$).

**Table 8 sensors-20-04435-t008:** Measured type B uncertainties of sampling and type A uncertainties of the OiW monitors’ confidence interval estimated in the calibration.

Volume Unit	Equipment	No. of Times	Volume [mL]	$u_{\mathrm{cB}}$ [ppm]	$u_{\mathrm{cA},\mathrm{OLS}}$ [ppm]	$u_{\mathrm{cA},\mathrm{WLS}}$ [ppm
$V_{Stock}$	$VR_{5}$	1	5.00	59.38	−	−
	$VR_{2}$	1	500			
$V_{0ppm}$	$VR_{1}$	10	10,000	-	1.92	0.03
$V_{5ppm}$	$VR_{5}$	1	5.00	0.04	1.87	0.04
$V_{10ppm}$	$VR_{5}$	1	5.00	0.09	1.83	0.09
	$VR_{6}$	1	0.01			
$V_{20ppm}$	$VR_{5}$	2	10.00	0.18	1.74	0.21
	$VR_{6}$	1	0.03			
$V_{40ppm}$	$VR_{5}$	4	20.00	0.35	1.59	0.44
	$VR_{6}$	2	0.12			
$V_{80ppm}$	$VR_{5}$	8	40.00	0.70	1.42	0.91
	$VR_{6}$	5	0.48			
$V_{160ppm}$	$VR_{4}$	1	81.96	1.31	1.74	1.86
$V_{320ppm}$	$VR_{3}$	1	167.98	2.54	3.60	3.74
$V_{400ppm}$	$VR_{4}$	1	86.09	3.15	4.68	4.68

**Table 9 sensors-20-04435-t009:** Combined and expanded uncertainty for each OiW concentration for both calibration methods: OLS and WLS.

Volume Unit	$u_{c,\mathrm{OLS}}$ [ppm]	$u_{c,\mathrm{WLS}}$ [ppm]	$U_{\mathrm{OLS}}$ [ppm]	$U_{\mathrm{WLS}}$ [ppm]
$V_{0ppm}$	1.92	0.03	3.76	0.06
$V_{5ppm}$	1.87	0.06	3.67	0.11
$V_{10ppm}$	1.83	0.13	3.59	0.25
$V_{20ppm}$	1.75	0.28	3.43	0.54
$V_{40ppm}$	1.63	0.56	3.19	1.10
$V_{80ppm}$	1.58	1.15	3.10	2.25
$V_{160ppm}$	2.18	2.28	4.28	4.47
$V_{320ppm}$	4.41	4.52	8.64	8.86
$V_{400ppm}$	5.65	5.65	11.07	11.07

**Table 10 sensors-20-04435-t010:** Data of the experiment containing the predicted concentration, each steady-state value of the four OiW monitors based on OLS calibration, and the grand mean of all four monitors for each concentration.

$X_{\mathrm{pred}}$ [ppm]	${\bar{C}}_{1}$ [ppm]	${\bar{C}}_{2}$ [ppm]	${\bar{C}}_{3}$ [ppm]	${\bar{C}}_{4}$ [ppm]	$\bar{\bar{C}}$ [ppm]
0	−0.45	−1.83	0.57	2.49	0.19
5	5.02	4.05	5.60	7.48	5.54
10	9.93	9.53	10.19	12.08	10.43
20	18.54	18.43	18.10	20.36	18.85
40	36.48	37.39	34.88	38.19	36.74
80	73.04	75.81	68.26	74.98	73.02
160	149.62	154.81	137.39	152.60	148.61
320	321.31	328.94	293.83	330.24	318.58
400	417.58	422.98	378.92	434.06	413.39

**Table 11 sensors-20-04435-t011:** Data of the experiment containing the predicted concentration, each steady-state value of the four OiW monitors based on WLS calibration, and the grand mean of all four monitors for each concentration.

$X_{\mathrm{pred}}$ [ppm]	${\bar{C}}_{1}$ [ppm]	${\bar{C}}_{2}$ [ppm]	${\bar{C}}_{3}$ [ppm]	${\bar{C}}_{4}$ [ppm]	$\bar{\bar{C}}$ [ppm]
0	−0.13	1.43	0.34	1.26	0.72
5	5.21	6.75	5.46	6.31	5.93
10	10.00	11.69	10.13	10.95	10.69
20	18.39	19.72	18.18	19.33	18.90
40	35.89	36.83	35.26	37.36	36.34
80	71.55	71.50	69.24	74.56	71.71
160	146.23	142.79	139.60	153.06	145.42
320	313.67	299.91	298.81	332.69	311.27
400	407.55	384.78	385.42	437.68	403.86

**Table 12 sensors-20-04435-t012:** Data of the results in Figure 15 and Figure 16, containing the predicted concentration, biggest deviation between all four OiW monitors for each concentration and each calibration method, and biggest deviation from $\bar{\bar{C}}$ for each concentration and each calibration method.

$X_{\mathrm{pred}}$ [ppm]	Biggest div. between ${\bar{C}}_{1}$, ${\bar{C}}_{2}$, ${\bar{C}}_{3}$, ${\bar{C}}_{4}$ with OLS [ppm]	Biggest div. between ${\bar{C}}_{1}$, ${\bar{C}}_{2}$, ${\bar{C}}_{3}$, ${\bar{C}}_{4}$ with WLS [ppm]	Biggest div. from ${\bar{\bar{C}}}_{\mathrm{OLS}}$ [ppm]	Biggest div. from ${\bar{\bar{C}}}_{\mathrm{OLS}}$ [%]	Biggest div. from ${\bar{\bar{C}}}_{\mathrm{WLS}}$ [ppm]	Biggest div. from ${\bar{\bar{C}}}_{\mathrm{WLS}}$ [%]
0	4.34	1.56	2.30	−	0.86	−
5	3.43	1.54	1.95	35.1	0.82	13.8
10	2.55	1.70	1.64	15.8	1.00	9.4
20	2.26	1.54	1.50	8.0	0.82	4.4
40	3.31	2.10	1.86	5.0	1.08	3.0
80	7.55	5.33	4.76	6.5	2.85	4.0
160	17.42	13.46	11.21	7.5	7.64	5.3
320	36.41	33.88	24.75	7.8	21.42	6.9
400	55.14	52.91	34.46	8.3	33.83	8.4

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
