# Peer review of "Uncertainty Analysis of Fluorescence-Based Oil-In-Water Monitors for Oil and Gas Produced Water"

_sensors, 2020, doi:10.3390/s20164435_

Round 1
Reviewer 1 Report
In this work, the authors evaluated four fluorescence-based monitors that are sensitive to the content of aromatic oil in a mixture. This study revealed that weighted least square (WLS) is better for higher reproducibility, and contrary to previous studies, it was found that measurements are not flow-dependent. For the contrary, other parameters were identified to being able to affect measurements.
This is indeed an interesting study that focus in the relevant measurement of oil-in-water concentration, which is a good fit for this journal. Moreover, this study is very complete.
Thus, my recommendation would be for acceptance if the length of the manuscript was so greatly exagerated. Namely, the main text must be reduced to at least half its current size, by taking advantage of the Supporting Information and by removing less important text.
Reviewer 2 Report
General Comment
The manuscript has conducted uncertainty analysis of fluorescence-based oil-in-water monitors for oil & gas produced water in a deoiling system. The uncertainty contributed from flow, gas bubbles, droplet sizes and quenchers were discussed. The topic is interesting and meaningful. However, the manuscript did not follow a constant logic that hard to read. For example, in Section 1, the authors first talked about the importance of continuous oil monitoring on PWRI. After that, it jumped to the practical drawbacks and impacts of the OiW monitors. Then, it jumped back to the importance of oil monitoring on produced water discharge. Section detailed introduced the uncertainty of the sampling and monitoring procedures and provided some calculations on how the uncertainty would fail the guideline. This part is extremely long, unsure whether it was a part of the introduction or discussion. Many parts have similar issues, such as separating experimental setups and experimental design… Significant efforts are needed to enhance the article in terms of precise and deliverable.
Therefore, it is not recommended for acceptance without a major revision.
Detailed Comments
Line 41-46, these two sentences were repeating each other that oil would increase the risk of clogging.
Line 181, the paragraph talking about the uncertainty of sampling, which was not addressed from the methodology to the discussion, please explain why it was important to the paper.
Line 268, the different composition of atomic structures, should be introduced in Section 2.
Line 288, why the abbreviation was “BLUE?” where did “L” come from?
Figure 2, it will be more apparent if the authors can change the dot plot to boxplot.
Line 795, can tap water represent the fluorescence sensitive matter in produced water?
Line 842, please explain how the results were influenced by the inner filter effects.
At what degrees the air bubbles, droplet size, and quencher have affected the uncertainty? Which one has the highest contribution?
Reviewer 3 Report
The paper by Dennis Severin Hansen et al. constitutes original article about the performance of four fluorescence-based oil-in-water monitors sensitive to the content of aromatic oil in a mixture. In my opinion the studies were performed properly. Various parameters characteristic for sensors like precision, repeatability, reproducibility were investigated. All parts of the manuscript are described plainly. From the obtained results Authors got appropriate conclusions. The paper is properly written - the quality of the text, figures and tables is rather good. In my opinion the manuscript merits to be published in Sensors journal in the present form.
1. Line 267: Authors claim that "The relative fluorescence intensity units (RFU) is then converted to ppm". I agree it can be, however only at very low concentrations. At higher concentrations there is a possibility that fluorescence intensity is not a linear function of a concentration. It should be taken into consideration and mentioned in the text. Secondly, there should be "are" instead of "is". "Units" is plural. 2. Line 437: Can Authors explain why gas bubbles reduce the strength of excitation and emission light? 3. Line 842: The sentence could be corrected. Inner filter effect in practice is not a quencher. The term quencher corresponds to a substance, which presence results in lowering a fluorescence intensity of a fluorophore. Inner filter effect in fact is a consequence of a presence of a quencher in the system.Author Response
Please see the attachment.

Round 2
Reviewer 1 Report
While I still believe that the manuscript is quite large, I do understand the need of the authors to provide a complete picture. Thus, my recommendation is for acceptance.
Author Response
Reviewer 1:
While I still believe that the manuscript is quite large, I do understand the need of the authors to provide a complete picture. Thus, my recommendation is for acceptance.
The authors have no further comments.
Reviewer 2 Report
1. Remove the grid from figure 2, 3, 12
Author Response
Reviewer 2:
1. Remove the grid from figure 2, 3, 12
Grid from Figure 2, 3, and 12 has been removed.